# DASC, a sensitive classifier for measuring discrete early stages in clathrin-mediated endocytosis

Xinxin Wang[1,2†], Zhiming Chen[2†], Marcel Mettlen[2], Jungsik Noh[1,2], Sandra L Schmid[2]*, Gaudenz Danuser[1,2]*

[1]Lyda Hill Department of Bioinformatics, University of Texas Southwestern Medical Center, Dallas, United States; [2]Department of Cell Biology, University of Texas Southwestern Medical Center, Dallas, United States

**Abstract** Clathrin-mediated endocytosis (CME) in mammalian cells is driven by resilient machinery that includes >70 endocytic accessory proteins (EAP). Accordingly, perturbation of individual EAPs often results in minor effects on biochemical measurements of CME, thus providing inconclusive/misleading information regarding EAP function. Live-cell imaging can detect earlier roles of EAPs preceding cargo internalization; however, this approach has been limited because unambiguously distinguishing abortive coats (ACs) from *bona fide* clathrin-coated pits (CCPs) is required but unaccomplished. Here, we develop a thermodynamics-inspired method, "disassembly asymmetry score classification (DASC)", that resolves ACs from CCPs based on single channel fluorescent movies. After extensive verification, we use DASC-resolved ACs and CCPs to quantify CME progression in 11 EAP knockdown conditions. We show that DASC is a sensitive detector of phenotypic variation in CCP dynamics that is uncorrelated to the variation in biochemical measurements of CME. Thus, DASC is an essential tool for uncovering EAP function.

*For correspondence:
sandra.schmid@utsouthwestern.edu (SLS);
gaudenz.Danuser@utsouthwestern.edu (GD)

†These authors contributed equally to this work

Competing interests: The authors declare that no competing interests exist.

## Introduction

Clathrin-mediated endocytosis (CME) is the major pathway for cellular uptake of macro-molecular cargo (*Kirchhausen et al., 2014*). It is accomplished by concentrating cell surface receptors into specialized 100–200 nm wide patches at the plasma membrane created by a scaffold of assembled clathrin triskelia (*Conner and Schmid, 2003*). The initiation and stabilization of these clathrin-coated pits (CCPs) is regulated by the AP2 (adaptor protein) complex (*Cocucci et al., 2012*), which recruits clathrin and binds to cargo and phosphatidylinositol-4,5-bisphosphate (PIP2) lipids. Numerous endocytic accessory proteins (EAPs), which modulate various aspects of CCP assembly and maturation, contribute to the formation of clathrin-coated vesicles (CCVs) that transport cargo to the cell interior. However, the exact functions of many of these EAPs are still poorly understood, and in some cases controversial (*Kaksonen and Roux, 2018*; *Mettlen et al., 2018*). Due to the resilience of CME, perturbing single EAPs, like CALM (*Xiao et al., 2012*; *Huang et al., 2004*), SNX9 (*Posor et al., 2013*; *Bendris et al., 2016*) etc., or even multiple EAPs (*Aguet et al., 2013*) often results in minor/uninterpretable changes in bulk biochemical measurements of cargo uptake. Nonetheless, perturbed EAP function can be physiologically consequential, *e.g.* mutations in CALM are linked to Alzheimer's disease (*Harold et al., 2009*) and SNX9 expression levels are correlated with cancer progression and other human diseases (*Bendris and Schmid, 2017*). These results bring into question whether measuring internalization by biochemical assays is sufficient for determining the actual phenotypes of missing EAP functions, and thereby further supporting clinical studies of the EAPs in more complex models.

Unlike bulk cargo uptake assays, the entire process of clathrin assembly and coated pit maturation at the plasma membrane can be monitored in situ by highly sensitive total internal reflection fluorescence microscopy (TIRFM) of cells expressing fiduciary markers for CCPs, such as the clathrin light chain fused to eGFP (*Mettlen and Danuser, 2014*). Using this imaging approach, we and others have observed that a large fraction of detected clathrin-coated structures (CSs) are shorter-lived (i.e. lifetimes < 20 seconds) than thought to be required for loading and internalizing cargo, and dimmer (i.e. exhibit lower intensities) than mature CCPs detected prior to internalization (*Taylor et al., 2011*; *Liu et al., 2010*). These so-called 'abortive' coats (ACs) presumably reflect variable success rates of initiation, stabilization and maturation; that is, the critical early stages of CME. However, the range of lifetimes and intensities of ACs overlaps substantially with the range of lifetimes and intensities of CCPs (*Figure 1A,B*). The current inability to unambiguously resolve ACs and CCPs limits analyses of the mechanisms governing CCP dynamics and their progression during CME.

Our initial attempts to solve this problem relied on a statistical approach to deconvolve the overall broad lifetime distribution of all detected CSs into subpopulations with distinct lifetime modes (*Loerke et al., 2009*). Although this method allowed the identification of three kinetically-distinct CS subpopulations (*Loerke et al., 2009*), the lifetimes of the thus identified subpopulations strongly overlapped, and the CS population with the longest average lifetimes, most likely representing productive CCPs, also contained a large fraction of very short-lived CCPs, which is structurally nonsensical. Later, as a result of improvements in the sensitivity of detection and tracking, eGFP-CLCa-labeled CSs were classified by imposing both lifetime and intensity thresholds (*Aguet et al., 2013*; *Kadlecova et al., 2017*). Besides the subjectivity in setting these critical values, we demonstrate here that neither lifetime nor intensity are sufficient to classify CSs. More recently, *Hong et al.*

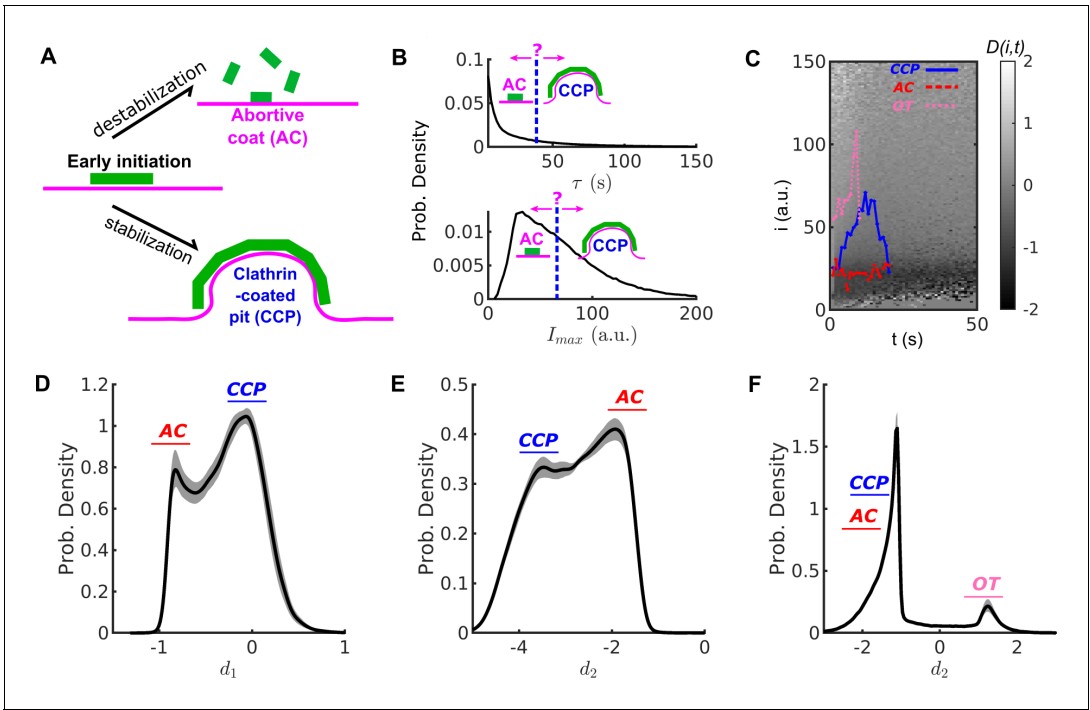

**Figure 1.** Conventional threshold-based cut-off vs. DAS derived metrics. (**A**) Schematic of abortive coat (AC) and clathrin-coated pit (CCP) evolving from early clathrin nucleation. (**B**) Lifetime ($\tau$) and intensity maxima ($I_{max}$) characteristics of hypothetical ACs and CCPs. ACs are typically assigned by a user-defined lifetime or $I_{max}$ threshold. (**C**) Disassembly risk map $D(i,t)$ represented on a gray value scale indicated by the gradient bar. A representative CCP (blue), AC (red) and outlier trace (OT) (pink) are plotted on the $D$-map. (**D**) Distribution of $N = 215,948$ counts of $d_1$ values for WT condition in black solid line. AC group near $d_1 < 0$ as a subpopulation, and CCP group at $d_1 \approx 0$ as another subpopulation. (**E**) Distribution of $N$ counts of $d_2$ values. Subpopulations of ACs and CCPs present in two modes. (**F**) Distribution of $N$ counts of $d_3$ values resolves the small subpopulation of OTs. Shaded area in (**D–E**) as 95% confidence intervals.

The online version of this article includes the following figure supplement(s) for figure 1:

**Figure supplement 1.** Fluctuation and heterogeneity in the intensity and $D$ traces.

*(2015)* removed some subjectivity by training a Support Vector Machine (SVM)-based classifier of 'false' vs 'authentic' CCPs; but the underlying features were still largely based on lifetime and intensity thresholds, which themselves are sensitive to detection and tracking artefacts (see *Aguet et al., 2013*). *Willy et al. (2017)* used growth rates to classify clathrin structures; however, the probability distribution of growth rate still shows strong overlaps between the proposed populations. Other efforts to distinguish abortive from productive events have introduced second markers, such as the recruitment of dynamin (*Grassart et al., 2014*; *Ehrlich et al., 2004*), or auxilin/GAK (*He et al., 2020*); however, these markers are also recruited to short-lived, potentially abortive CCPs, especially when exogenously expressed (*Ehrlich et al., 2004*; *Massol et al., 2006*; *Taylor et al., 2012*). Perhaps the most unambiguous means of identifying productive CCPs is by the internalization of pH sensitive-cargo (*Taylor et al., 2011*), with the obvious drawback of dual labeling and a more complicated experimental set up. Clearly, the mechanistic analysis of CCP dynamics would greatly benefit from an objective and unbiased means to resolve these heterogeneous subpopulations.

Here, we introduce a thermodynamics-inspired method, referred to as *disassembly asymmetry score classification* (DASC), that resolves ACs from CCPs relying on the differential asymmetry in frame-by-frame intensity changes between disassembling and fluctuating/growing structures. DASC is independent of user-defined thresholds and prior assumptions, and does not require second markers. We confirmed the positive correlation between CCP stabilization and curvature generation by combining DASC with quantitative live cell TIRF and epifluorescence microscopy. We further applied DASC to phenotype siRNA-mediated knockdown of eleven reportedly early-acting and 'pioneer' EAPs on CCP initiation and stabilization and measured their effects on CS dynamics and on cargo uptake. In most cases we detected significant effects on early stages of CME resulting from reduced CCP initiation and/or stabilization that did not correlate with changes in transferrin uptake. Thus, DASC reveals EAP functions that are not detected by traditional bulk biochemical measurements. Together these studies establish DASC as a unique tool for objectively distinguishing abortive coats from *bona fide* CCPs and thus indispensable for comprehensively revealing which EAPs act at specific stages to mediate endocytic coated vesicle formation.

## Results

### Disassembly Asymmetry Score Classification (DASC): a new method to analyze CCP growth and stabilization

To ensure high sensitivity detection of all CCP initiation events, ARPE19/HPV16 (hereafter called HPV-RPE) cells were infected with lentivirus encoding an eGFP-tagged clathrin light chain a (eGFP-CLCa) and then selected for those that stably expressed eGFP-CLCa at ~ 5-fold over endogenous levels. Overexpression of eGFP-CLCa ensures near stoichiometric incorporation of fluorescently-labeled CLC into clathrin triskelia by displacing both endogenous CLCa and CLCb. Control experiments by numerous labs have established that under these conditions CME is unperturbed and that eGFP-CLCa serves as a robust fiduciary marker for coated pit dynamics at the plasma membrane (*Aguet et al., 2013*; *Taylor et al., 2011*; *Loerke et al., 2009*; *Ehrlich et al., 2004*; *Miller et al., 2015*; *Gaidarov et al., 1999*; *Cocucci et al., 2012*). For all conditions, ≥ 19 independent movies were collected and the eGFP intensities of >200,000 clathrin structures per condition were tracked over time using TIRFM and established automated image analysis pipelines (*Aguet et al., 2013*; *Jaqaman et al., 2008*). We refer to these time dependent intensities as traces. Each trace is a measure of the initiation, growth and maturation of the underlying clathrin structure (CS).

Following their initiation, the detected CSs are highly heterogeneous, reflected by the widely spread distributions of lifetime and intensity maxima of their traces (*Figure 1B*, top and bottom panels, respectively). Previous studies (*Aguet et al., 2013*; *Loerke et al., 2009*; *Ehrlich et al., 2004*) have suggested that this heterogeneity reflects a mixture of at least two types of structures: 1) stabilized, *bona fide* CCPs, and 2) unstable partial and/or abortive coats (ACs) that rapidly turnover.

Bona fide CCPs tend to have lifetimes > 20 s and approach an intensity level corresponding to a fully assembled coat (between 36 and > 60 triskelia) (*Ehrlich et al., 2004*). In contrast, ACs tend to exhibit lower intensity levels and disassemble at any time. However, CCPs and ACs strongly overlap in their lifetime and intensity distributions, especially during the critical first 20–30 s after initiation. Consequently, the contributions of these two functionally distinct subpopulations of CSs to the

overall lifetime or intensity distributions cannot be resolved and ACs cannot be readily distinguished from CCPs by application of a lifetime or intensity threshold (*Figure 1B*). Significantly compounding the ability to distinguish CCPs from ACs is the fact that the intensities of individual CSs are highly fluctuating (see for example, *Figure 1—figure supplement 1A–B*). These fluctuations are inevitable and reflect a combination of low signal:noise (especially during early stages of CCP growth and maturation), rapid turnover of individual triskelia, which occurs on the time scale of ~2 s (*Kirchhausen et al., 2014*; *Avinoam et al., 2015*), stochastic bleaching of fluorophores, camera noise and membrane fluctuations within the TIRF field. We thus sought an approach to distinguish ACs from *bona fide* CCPs that is independent of user-defined thresholds and leverages these intensity fluctuations measured at high temporal resolution.

Inspired by the computation of entropy production (EP) (*Seifert, 2005*), we designed a new metric derived from the fluctuations of clathrin intensity traces that can clearly separate ACs from CCPs. Conventionally, EP quantifies the dissipation rate of thermal energy when a system of interest is driven far away from equilibrium, as is the case during the formation of a macro-molecular assembly such as a CCP. This quantity is obtained by computing the difference between forward and reverse reaction rates. We therefore assigned clathrin assembly and disassembly as forward and reverse reactions in order to derive an EP-based metric of the progression of CS formation.

We first expressed each trace as a chain of transitions among integer intensities (or states) over time, for the nth trace,

$$I_n(t) := (i, t = 1s) \rightarrow (j, t = 2s) \rightarrow \ldots \rightarrow (k, t = \tau) \qquad (1)$$

In this example, $i, j \ldots k \in [1, i_{max}] (a.u.)$, where $i_{max}$ is the largest intensity recorded so that $[1, i_{max}]$ represents the entire pool of the intensity states. $\tau$ is the lifetime of this trace (see Materials and methods for details).

Next, after expressing all the traces as in Equation 1, we quantified for each transition between two intensity states the conditional probabilities $W_t(i^-|i)$ and $W_t(i|i^-)$. Given state $i$ and its **lower states** $i^- \in [1, i-1]$, $W_t(i^-|i)$ denotes the probability of a decrease in intensity $i \rightarrow i^-$ between time $t$ to $t+1$, and $W_t(i|i^-)$ denotes the probability for an increase in intensity $i^- \rightarrow i$ (see Materials and methods for details).

From these probabilities, we define a disassembly risk function ($D$) for any given intensity-time state $(i, t)$ as:

$$D(i, t) = ln \frac{\sum_{i^-=1}^{i-1} W_t(i^-|i)}{\sum_{i^-=1}^{i-1} W_t(i|i^-)} = ln \underbrace{\sum_{i^-=1}^{i-1} W_t(i^-|i)}_{①} - ln \underbrace{\sum_{i^-=1}^{i-1} W_t(i|i^-)}_{②} \qquad (2)$$

where, between state $i$ and its lower states $i^-$ at time $t$, Term ① includes every transition of *clathrin loss*; while Term ② includes every transition of *clathrin gain*. D = ① - ② thus indicates the net risk for disassembly at every intensity-time state.

We can use this $D$ function to project each trace into a space of disassembly risk (*Figure 1C*). The projected trace (*Figure 1—figure supplement 1C*) then predicts the disassembly risk for an individual trace of particular intensity at a specific time. For example, $I_n(t)$ in *Equation 1* yields a corresponding series of disassembly risk (see *Figure 1—figure supplement 1C*), written as:

$$D[I_n(t), t] = D(i, t = 1) \rightarrow D(j, t = 2) \rightarrow \ldots \rightarrow D(k, t = \tau) \qquad (3)$$

Hence, each intensity trace as in *Equation 1* is translated into a $D$ series reflecting the risk of disassembly at each time point. Most $D(i, t)$ values are either negative (low disassembly risk, that is loss < gain) or nearly zero (moderate disassembly risk), see *Figure 1C*, which we interpret as reflective of two phases of CCP growth and maturation.

1. Early growth phase: Following an initiation event, and during the first few seconds of CME, almost all CSs, including ACs, grow albeit with fluctuation. Also, most CSs are still small. Hence, in this earliest phase, Term ① < Term ② and $D(i, t) < 0$. Accordingly, clathrin dissociation is rare and all traces in this early phase have a low risk of disassembly. However, the risk of acute disassembly increases as CSs approach the end of this phase. CSs that disassemble early are potentially ACs, whereas surviving CSs enter the next phase to become CCPs.

2. Maturation phase: Upon completion of the growth phase, CCP intensities plateau but continue to fluctuate over many high intensity states at mid to late time points. The fluctuation is equivalent to having a similar chance of gain or loss of clathrin, thus Term ① ≈ ② and $D(i,t) ≈ 0$. CCPs in this phase retain a moderate risk of acute disassembly.

In summary, $D(i,t)<0$ is indicative of early stages of clathrin recruitment when disassembly risk is suppressed; $D(i,t) = 0$ is indicative of intensity fluctuations that occur at later stages of CCP growth and maturation. *Figure 1C* displays representative examples of CCP (blue) and AC (red) traces. In the early growth phase, both traces exhibit $D(i,t)<0$ (dark shaded background). As CCPs reach the maturation phase they approach the regime $D(i,t) ≈ 0$. At early time points, both ACs and CCPs exist in the early growth phase ($D(i,t)<0$); however, with time, only CCPs enter the maturation phase ($D(i,t) ≈ 0$), and accordingly $D$ values increase. Thus, for maturing CCPs $D$ values distribute around zero, whereas for ACs $D$ values distribute in the negative range.

A small portion of CSs possess abnormally high intensities when first detected, but quickly disappear. Therefore, Term ① > Term ②, $D(i,t)>0$, and the disassembly risk for high intensity states at early time points is high (pink traces in *Figure 1C* and *Figure 1—figure supplement 1C*). These atypical CSs frequently appear in regions of high background, which can obscure early and late detections (*Figure 1—figure supplement 1D*) and impair the ability to accurately detect small intensity fluctuations. As interpreting the fates of these CSs is difficult, and because they are rare, we refer to them as outlier traces (OTs).

To quantitatively distinguish the distributions of CCPs and ACs, we examined mean, variation and skewness of the $D$ series. Considering the $n$th series $D[I_n(t),t]$, we first calculated its time average:

$$d_1(n) = \frac{1}{\tau}\sum_{t=1}^{\tau} D[I_n(t),t]$$

An AC is expected to have $d_1(n)<0$, whereas a CCP is expected to have $d_1(n) ≈ 0$. Indeed, for a population of $N > 200,000$ CSs tracked in HPV-RPE cells, the distribution of $d_1$ values is bimodal (*Figure 1D*), allowing the distinction of ACs and CCPs.

We additionally computed:

$$d_2(n) = \ln\{[\max(D[I_n(t),t]) - \min(D[I_n(t),t])]/\hat{\tau}\}$$

which reflects the lifetime-normalized difference between the maximum and minimum value of a $D$ series (dimensionless lifetime $\hat{\tau} = \tau/1s$ is used). Note that by definition, the first detection gives $D = 0$ for both ACs and CCPs (see *Figure 1—figure supplement 1C*); thus, the maximum $D$ value for ACs is always 0. For example, the $D$-series of the CCP trace in *Figure 1C* (see blue curve in *Figure 1—figure supplement 1C*) has a maximum value of 0.2 and minimum value of -0.8, and lasts for 30s. Thus $d_2 = \ln\{[0.2 - (-0.8)]/30\} ≈ -3.4$. The $D$-series of the AC trace (red curve in *Figure 1—figure supplement 1C*) has a maximum value of 0 and a minimum value of -1, yielding $d_2 = \ln\{[0 - (-1)]/10\} ≈ -2.3$. The distribution of this feature is also bimodal (*Figure 1E*) and thus can strengthen the distinction between ACs and CCPs.

The $D$ series of OTs contain a few initial values that are much higher than those in the $D$ series associated with either ACs or CCPs (*Figure 1—figure supplement 1C*). Therefore, such series can be identified via a modified skewness of $D$:

$$d_3(n) = \frac{1}{\tau}\sum_{t=1}^{\tau}\frac{[D[I_n(t),t] - d_1(n)]^3}{\sigma(n)^3},$$

where $\sigma(n) = \sqrt{\frac{1}{\tau}\sum_{t=1}^{\tau}[D[I_n(t),t] - d_1(n)]^2}$ is the standard deviation of the $D$ series.

The distribution of $d_3$ over $N$ series displays two tight populations with the $d_3$ values of OTs easily separable from the $d_3$ values of ACs and CCPs (*Figure 1F*).

Using the three summary statistics $(d_1, d_2, d_3)$ we project all CS traces into a feature space (*Figure 2A*) and classify ACs (red), CCPs (blue), and OTs (pink) using k-medoids clustering (see Materials and methods). Values for $d_3$ identify OTs, whereas $d_1$ and $d_2$ complement one another separating ACs from CCPs. As these features originate from the disproportionate disassembly vs. assembly of CSs, we term our feature selection the *disassembly asymmetry score* (DAS), and name

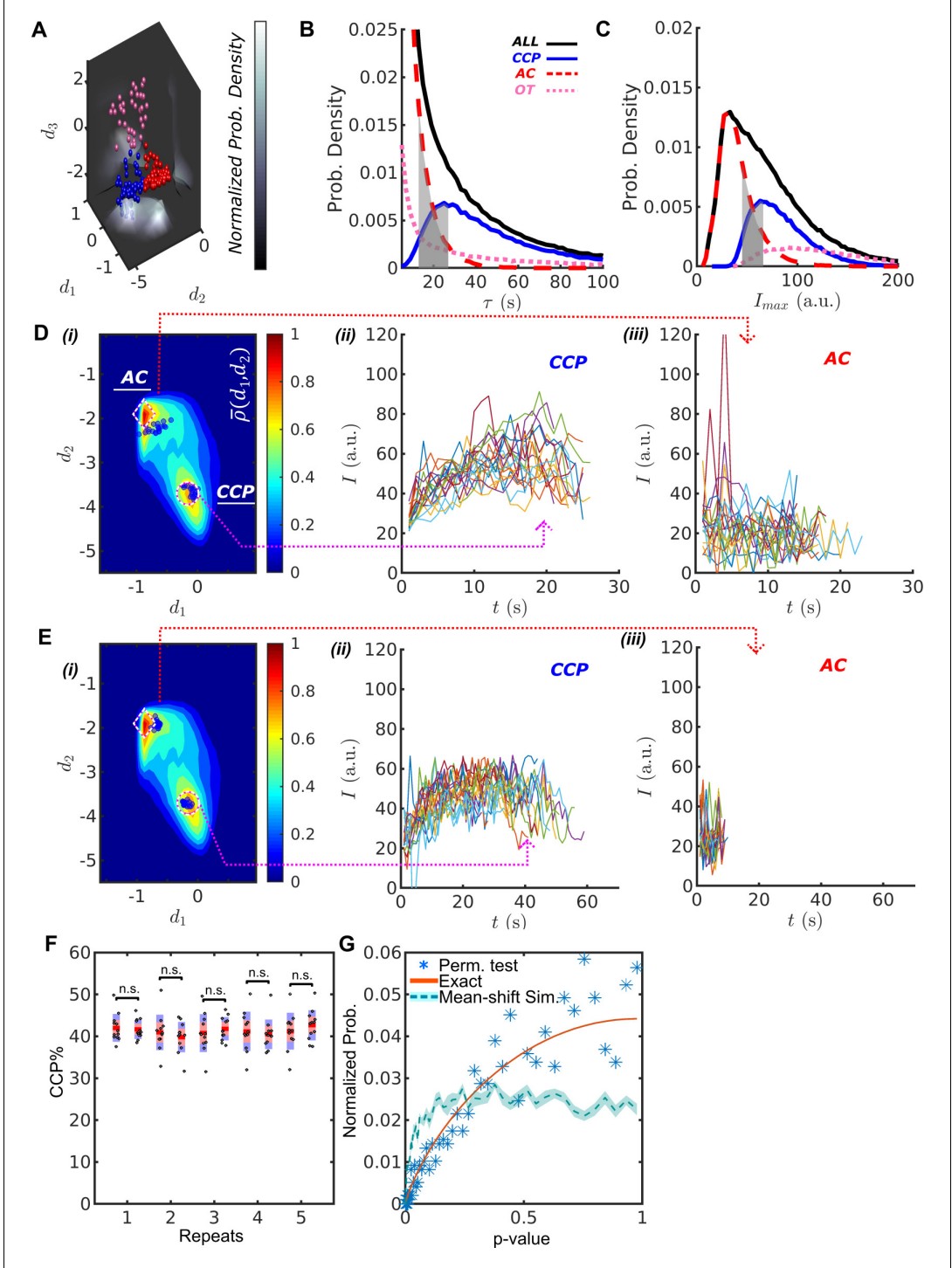

**Figure 2.** DASC resolves behaviorally distinct ACs and CCPs. (A) k-medoid classification in three-dimensional feature space $(d_1, d_2, d_3)$, where normalized probability densities $\bar{\rho}(d_1, d_2)$, $\bar{\rho}(d_2, d_3)$ and $\bar{\rho}(d_1, d_3)$ are shown as three landscape plots. $\bar{\rho}$ values are scaled according to gray bar. Examples of CCPs (blue), ACs (red) and OTs (pink) concentrate near the maxima of $\bar{\rho}$. (B) Lifetime distributions of all CCPs ($n_{CCP}$ traces in blue/solid), ACs ($n_{AC}$ traces in red/dashed), OTs ($n_{OT}$ traces in pink/dotted) and all traces ($n_{CCP} + n_{AC} + n_{OT}$ traces in black/solid). Gray region shows lifetime overlap between CCPs and ACs. (C) $I_{max}$ distributions and overlap. Color scheme same as in (B). Gray region shows $I_{max}$ overlap between CCPs and ACs. (D) i. DAS plot: $\bar{\rho}(d_1, d_2)$ contour map (values indicated by 'rainbow' color bar) with modes for CCPs and ACs indicated by circle and diamond, respectively. Ten representative CCPs and ACs (blue dots) from the lifetime overlap in (B) close to the modes are projected onto $d_1$-$d_2$ coordinate. Traces of the representative CCPs (ii) and ACs (iii) from i. (E) Same as (D) for the representative CCPs and ACs from the $I_{max}$ overlap in (C). (F) Five repeats of

*Figure 2 continued on next page*

*Figure 2 continued*

comparing $CCP\% = n_{CCP}/(n_{CCP} + n_{AC} + n_{OT}) \times 100\%$ between 12 and another 12 movies of siControl cells imaged on the same day. A total of 24 movies were randomly shuffled to obtained 12-12 pairs. (G) Normalized probability of given p-values using Wilcoxon rank sum test. 1. Blue asters: 1000 repeats of the same shuffle in (F); 2. Red solid line: exact solution to 12-12 random sample test; 3. Light-blue dashed line: simulation of 1000 tests between 12 normally distributed random numbers (NRN) with mean $\mu = 0$ and standard deviation $\sigma = 1$ and another 12 NRNs with $\mu = 0.5$ and $\sigma = 1$. Shaded area as 95% confidence interval obtained from 10 repeated simulations.

The online version of this article includes the following figure supplement(s) for figure 2:

**Figure supplement 1.** 'Elbow' method for estimating optimal cluster number.

**Figure supplement 2.** Ambiguity of traces at the hard boundary between AC and CCP populations generated by k-medoids clustering.

the packaged software DASC as DAS classification, available under https://github.com/DanuserLab/cmeAnalysis (*Wang et al., 2020*; copy archived at https://github.com/elifesciences-publications/cmeAnalysis).

## DASC accurately identifies dynamically distinct CS subpopulations

The DASC-resolved subpopulations of CSs exhibit distinct but overlapping lifetimes and intensities (*Figure 2B,C* grey zone), confirming the inability of these conventional metrics to distinguish ACs from CCPs. The lifetimes of ACs (*Figure 2B*, red) were predominantly short (<20 s) and exhibited an exponential distribution characteristic of coats that are exposed to an unregulated disassembly process. In contrast, CCP lifetimes (*Figure 2B*, blue) exhibited a unimodal distribution with a highest probability lifetime of ~26 s. In previous work, we had shown that this distribution is best represented by a Rayleigh distribution that reflects the kinetics of a three- to four-step maturation process (*Aguet et al., 2013*). Interestingly, although partially overlapping with ACs, the intensity distribution of CCPs (*Figure 2C*) exhibits a sharp threshold in the minimum intensity, as would be expected given the minimum number of clathrin triskelia required to form a complete clathrin basket (*Grassart et al., 2014*; *Ehrlich et al., 2004*). The majority of OTs (*Figure 2B,C*, pink) are highly transient and bright structures and hence unlikely to be functionally relevant clathrin assemblies.

Despite their overlapping lifetimes and intensities, AC and CCP traces are well resolved by DASC as represented in two-dimensional, normalized probability density maps $\bar{\rho}(d_1, d_2)$ (*Figure 2Di*), from here on referred to as DAS plots (see Materials and methods for details). To illustrate this point, we selected 10 CSs with overlapping lifetime distributions (10-25 s, gray zone *Figure 2B*) that fall close to the associated modes of either the AC or CCP populations in the DAS plot, that is the two maxima of $\bar{\rho}(d_1, d_2)$ denoted by a diamond for ACs and circle for CCPs in Fig. 2Di. Blue dots show the $(d_1, d_2)$ locations of the selected CSs. Their intensity traces are shown in *Figure 2Dii-iii*. Although the lifetimes are almost identical, the CCP and AC traces show characteristic differences in their intensity evolution. CCP intensities rise to a clear maximum as they assemble a complete clathrin coat (*Figure 2Dii*), followed by a falling limb, which is associated with CCV internalization and/or uncoating. In contrast, AC intensities are lower and more random (*Figure 2Diii*), suggesting that these coats, trapped in the early growth phase, undergo continuous exchange of clathrin subunits without significant net assembly. We occasionally observed rapidly fluctuating, high intensity CSs amongst the AC traces. These likely correspond to previously identified 'visitors' (i.e. endosome-associated coats transiting through the TIRF field), which make up ~10% of all detected CSs (*Aguet et al., 2013*).

We next selected 10 CSs (indicated as blue dots in *Figure 2Ei*) from the AC and CCP populations that fall into the overlap region in the distributions of intensity maxima (gray zone, *Figure 2C*). Although the intensity ranges are nearly identical, the selected CCP traces again display a rising and falling limb and lifetimes of ~60 s (*Figure 2Eii*). In sharp contrast, ACs fluctuate about the same intensity values (*Figure 2Eiii*) and exhibit much shorter lifetimes of ~10 s.

To establish the statistical robustness and reproducibility of the DASC-based metrics, we acquired 24 movies from the same WT condition on the same day and randomly separated them into pairs of 12 movies each. We then applied DASC to the movies, and calculated percent contribution of *bona fide* CCPs, $CCP\% = n_{CCP}/(n_{CCP} + n_{AC} + n_{OT}) \times 100\%$ (where $n$ indicates population), for each movie using the first 12 as the control set, the other 12 as the test set, and compared the two data sets. This permutation test was repeated 1000 times. *Figure 2F* shows 5 example pairs

indicating no significant difference between samples (statistical significance obtained using Wilcoxon rank sum method). Then, we calculated p-values, $p$, from all repeats and presented the values as normalized probabilities $P(p)$ (blue asters in *Figure 2G*). Note that the number of possible p-values resulting from the small sample 12-12 Wilcoxon rank sum is finite. Therefore, $P(p)$ equals to a histogram of discrete p-values. The expected p-value histogram for a 12-12 Wilcoxon rank sum test of random samples is presented as a reference curve (red). The data follows this curve closely. In contrast, a comparison between Gaussian random numbers with different means led to a significant shift from the reference curve (light blue, dashed line). These test results confirmed that no significant difference exists between movies from the same condition. Thus, DASC is statistically robust and not overly sensitive to movie-to-movie variations in data collected on the same day.

Together, these data demonstrate that DASC is a robust tool to discriminate between two completely distinct clathrin coat assembly and disassembly processes that, by inference, are associated with abortive coats and *bona fide* CCPs. This has not been possible based on more conventional features such as lifetime and intensity (*Aguet et al., 2013*; *Loerke et al., 2009*; *Kadlecova et al., 2017*; *Hong et al., 2015*; *Ehrlich et al., 2004*; *Bucher et al., 2018*).

## DASC detects 3 populations of CSs in multiple cell types

The qualitative inspection of the $d_{1,2,3}$-feature spread follows three modes; hence we initially assigned $k = 3$ clusters to distinguish AC, CCP, and OT populations. Although previous work also suggested three CS subpopulations (*Aguet et al., 2013*; *Loerke et al., 2009*), we wished to validate this parameter selection more broadly. Besides HPV-RPE we applied DASC analysis to several other cell lines expressing eGFP or mRuby-CLCa, including ARPE, SK-Mel-2 (SKML), H1299 and A549. After calculating the three DAS features $d_{1,2,3}$, we applied k-medoids clustering for various cluster number $k = 1 \ldots 7$ in these cells. The total distance from all the traces to their host clusters' centers in the $d_{1,2,3}$ space is calculated for each $k$. We observed that for all cell lines tested, except for A549, a clear "elbow" appears at $k = 3$ (see *Figure 2—figure supplement 1*), indicating the optimal cluster/population number is consistently 3 (*Kodinariya and Makwana, 2013*). As previously described for cmeAnalysis (*Aguet et al., 2013*; *Mettlen and Danuser, 2014*), the signal:noise in cells expressing genome-edited CLCa was unsuitable for DASC analysis (not shown). Nor could we apply DASC to cells with large fractions of static CCPs (e.g. A549 cells, *Figure 2—figure supplement 1 ii*). Nonetheless, from this we conclude that DASC is reliably transferable to multiple cell lines exhibiting dynamic, diffraction-limited CSs.

To determine how sharp the boundaries are between the k-medoids generated AC and CCP populations, we identified the 10% traces that locate at equal distances to the AC and CCP medoid (*Figure 2—figure supplement 2A*). As expected for a clustering of continuously distributed data, the lifetime and maximal intensity histograms of these edge traces fell in between the corresponding distributions of AC and CCP (*Figure 2—figure supplement 2B,C*). Also, the intensity cohorts show that CCPs and edge traces are similar in dynamics, although CCPs have higher intensities (*Figure 2—figure supplement 2D*). However, this ambiguity of edge traces has no effect on DASC's ability to resolve functionally, structurally and kinetically distinct AC and CCP populations. We confirmed this by testing whether the exclusion of 10% edge traces would change the phenotype of a molecular perturbation such as CALM KD. Focusing on the CCP% as a readout of the phenotype, elimination of edge traces slightly reduced the values for both siControl and siCALM (*Figure 2—figure supplement 2E*), but the relative difference in CCP% between the two conditions remained the same. Thus, conclusions drawn from DASC on shifts in AC and CCP populations are robust, even though the AC and CCP populations per se are not fully separable.

## Validation through perturbation of established CCP initiation and stabilization pathways

We next tested the performance of DASC against conditions known to perturb early stages in CME. AP2 complexes recruit clathrin to the plasma membrane and undergo a series of allosterically-regulated conformational changes needed to stabilize nascent CCPs (*Kadlecova et al., 2017*; *Jackson et al., 2010*; *Kelly et al., 2008*; *Collins et al., 2002*). Previous studies based on siRNA-mediated knockdown (KD) of the α subunit of AP2 and reconstitution with either WT, designated αAP2(WT), or a mutant defective in PIP2 binding, designated αAP2(PIP2⁻), in hTERT-RPE cells have

established that αAP2-PIP2 interactions are critical mediators of AP2 activity (*Kadlecova et al., 2017*). We repeated these experiments in HPV-RPE cells using DASC and detected pronounced differences in the DAS plots derived from αAP2(WT) vs αAP2(PIP2⁻) cells (*Figure 3Ai–ii*). A DAS difference $\rho[\alpha AP2(WT), \alpha AP2(PIP2^-)]$ map (see Materials and methods) shows a dramatic increase (yellow) in the fraction of ACs and a corresponding decrease (black) in the fraction of CCPs (*Figure 3B*), as expected given the known role of αAP2-PIP2 interactions in CCP stabilization.

We also observed an increase in CS initiation rate (CS init.) (*Figure 3Ci*), measured by total trackable CSs detected per minute per cell surface area (see Material and methods for detail definition). Previous studies reported a decrease in CS initiation rate (*Kadlecova et al., 2017*). This apparent discrepancy likely reflects our use of all detected traces to calculate CS initiation rate as compared to previous use of only 'valid' tracks (see Materials and methods). As the CSs observed in the αAP2-PIP2⁻ cells were significantly dimmer than those detected in WT cells (see *Figure 3—figure supplement 1*), more initiation events would have been scored as 'invalid' in the previous analysis due to flawed detections, especially at early stages of CCP assembly.

DASC analysis revealed multiple defects in early stages of CME in the αAP2(PIP2⁻) cells compared to αAP2(WT) cells. Note that the k-medoids clustering was determined based on control movies and then the boundaries between control populations were applied to the traces extracted from movies with experimental perturbations. We detected a pronounced decrease in the efficiency of CCP stabilization, measured as CCP% (*Figure 3Cii*), which was calculated as the fraction of CCPs in all the valid traces (see Material and methods). The AC% increased proportionally to the decrease in CCP% as expected (data not shown). The lifetime distributions of CCPs also shifted to shorter lifetimes (*Figure 3Ciii*, left panel), resulting in decreased median lifetimes (*Table 1*) in αAP2(PIP2⁻) cells compared to αAP2(WT) cells. This lifetime shift indicates that the mutation can also cause instability in fully grown clathrin coats, as previously reported (*Kadlecova et al., 2017*). There was no change in lifetime distribution of ACs (*Figure 3Ciii*, right panel).

We next compared the kinetics and extent of recruitment of AP2 and clathrin to ACs and CCPs. For this we applied two-color imaging and 'master-slave' analysis (*Aguet et al., 2013*) to simultaneously track clathrin and AP2 in ARPE cells stably expressing mRuby2-CLC as the master channel and the wild-type α subunit of AP2 encoding eGFP within its flexible linker region (α-eGFP-AP2) as the slave channel. For this and our further comparisons, we focused on traces with lifetimes in the range 15 s to 25 s (*Loerke et al., 2011*) as these corresponded to the maximum overlap between ACs and CCPs (grey zone, *Figure 2B*). Applying DASC to the mRuby2-CLCa signal to distinguish CCPs from ACs we observed, as expected, that CCPs reach significantly higher average clathrin intensity than ACs (*Figure 3Di–ii*, additional 5–15 s and 25–35 s cohorts are shown in *Figure 3—figure supplement 2*). We also observed significantly higher levels of AP2 α subunit, relative to clathrin, present at CCPs than ACs. Moreover, as previously shown for statistically-defined abortive vs. productive pits (*Loerke et al., 2011*), the initial rates of recruitment to CSs of both clathrin and AP2, determined by the derivative of intensity, were much greater for CCPs than ACs (*Figure 3Ei–ii*). Together these data corroborate the known stabilization function of AP2 during CCP initiation (*Kadlecova et al., 2017*; *Owen et al., 2004*), and serve to validate the ability of DASC to distinguish different regimes of molecular regulation at early stages of CME.

## Validation through curvature acquisition and CCP stabilization

Previous studies have suggested that curvature generation within nascent CCPs is a critical factor for their maturation and that CCPs that fail to gain curvature are aborted (*Aguet et al., 2013*; *Loerke et al., 2009*; *Bucher et al., 2018*; *Mettlen et al., 2009*). Therefore, we compared the acquisition of curvature in DASC-identified ACs and CCPs (*Figure 3F*). To this end, we applied DASC to traces acquired by near simultaneous epifluorescence (EPI)-TIRF microscopy (*Aguet et al., 2013*; *Loerke et al., 2009*). Because of the differential fluorescence excitation depths of TIRF- and epi-illumination fields, the ratio of EPI:TIRF intensities of individual CSs provides a measure of curvature (*Figure 3—figure supplement 3A*). CSs were classified as ACs or CCPs based on the TIRF channel traces and then grouped into lifetime cohorts to obtain average invagination depth $\Delta z$ (See Materials and methods and materials). DASC-defined CCPs in the 20s cohort reached maxima $\Delta z_{max}/h = \max[\Delta z(t)]/h > 0.3$ (*Figure 3F*), which corresponds to an invagination depth of $>35nm$ ($h = 115nm$ is the characteristic depth of our TIRF illumination field, see Materials and methods). In contrast, DASC-defined ACs in the 20 s cohort fail to gain significant curvature. Other cohorts

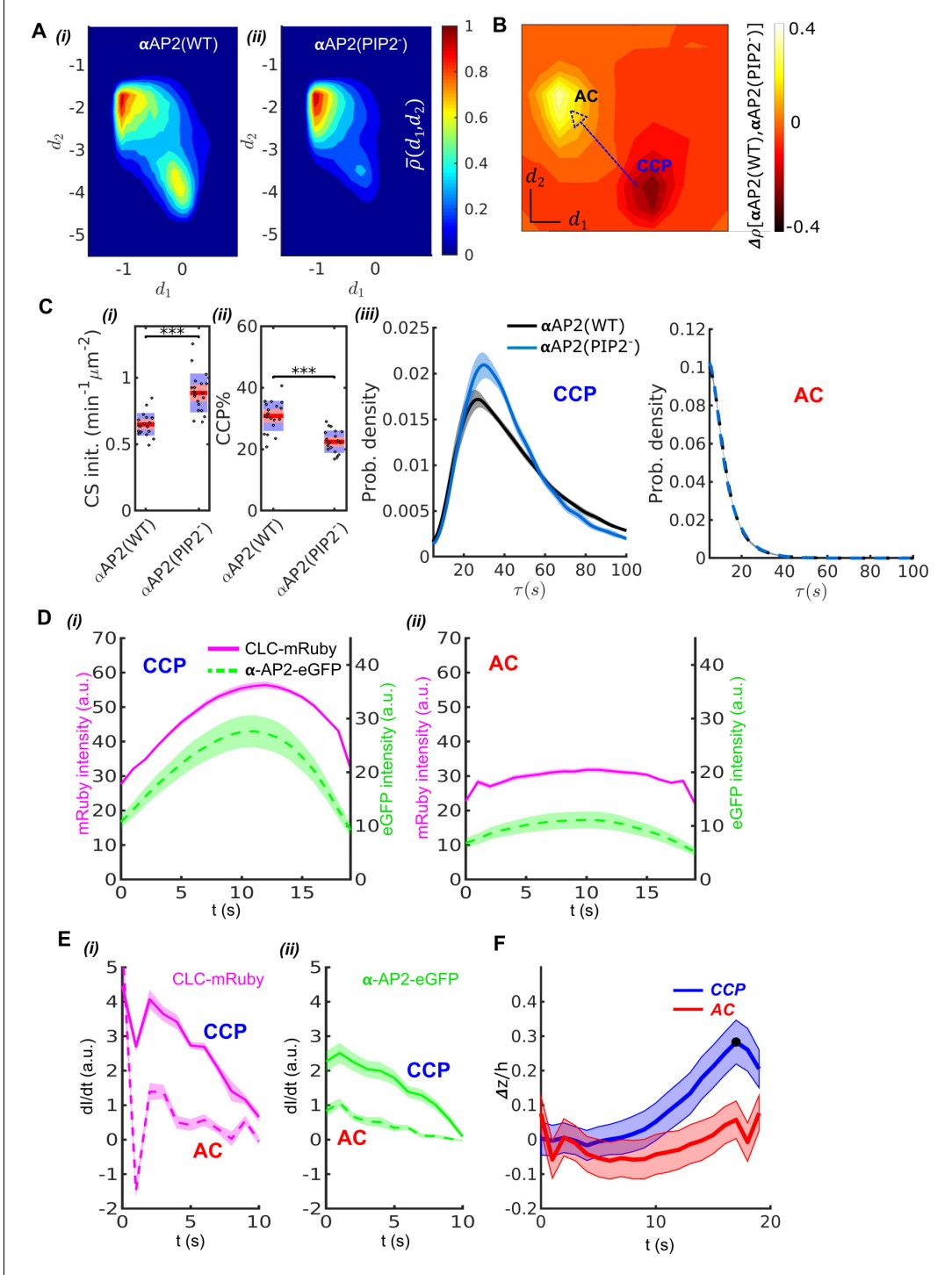

**Figure 3.** Validation of DASC. (A) DAS plots showing $\bar{\rho}(d_1, d_2)$ contour as 'rainbow' color map and color bar for αAP2(WT) cells (i) and αAP2(PIP2⁻) cells (ii). (B) DAS difference plot (difference in $d_1 d_2$ distribution) of αAP2(PIP2⁻) minus αAP2(WT) cells as contour in 'heat' map. (C) Comparison of DASC-derived metrics for CCP dynamics in αAP2(WT) vs αAP2(PIP2⁻) cells, (i) CS initiation rate and (ii) CCP% in αAP2(WT) and αAP2(PIP2⁻) cells. Dots represent jittered raw data from individual movies, box plots show mean as red line and 95% and 1 standard deviation as red and blue blocks, respectively (see Materials and methods). (iii) CCP lifetime distribution of αAP2(WT) vs αAP2(PIP2⁻) cells. (iv) $I_{max}$ distribution of ACs in αAP2(WT) vs αAP2(PIP2⁻) cells. (D) 20 second cohorts from dual channel movies of CLC-mRuby (magenta, solid) and α-AP2-eGFP (green, dashed) for CCPs (i) and ACs (ii). (E) Time derivative of CLC-mRuby (i) and α-AP2-eGFP (ii) intensities for the first 10 seconds in the dual channel cohorts of CCPs and ACs in (D). (F) Time course of invagination depth $\Delta z(t)/h$ (TIRF characteristic depth $h = 115nm$) for CCPs (blue) and ACs (red) measured by Epi-TIRF. Statistical

*Figure 3 continued on next page*

*Figure 3 continued*

analysis of the data used the Wilcoxon rank sum test. *** p-value < 0.001, ** p-value < 0.01, * p-value < 0.05, n.s. (non-significant) p-value > 0.05. Shaded area indicates 95% confidence interval for all plots.

The online version of this article includes the following figure supplement(s) for figure 3:

**Figure supplement 1.** Single frame from movies of αAP2(WT) and αAP2(PIP2⁻) cells.
**Figure supplement 2.** 10 s (i–ii) and 30 s (iii–iv) cohorts of CCPs and ACs obtained from CLC-mRuby and α-AP2-eGFP movies.
**Figure supplement 3.** DASC combined with EPI-TIRF approach reveals CME invagination kinetics.

supporting this CCP-curvature relation are presented in *Figure 3—figure supplement 3B-D*. Importantly, the relationship is conserved even when we analyzed a subset of ACs and CCPs that exhibit the same range of maximal intensities (*Figure 3—figure supplement 3E-G*). Together, these data establish that DASC-resolved ACs and CCPs are, indeed, structurally and functionally distinct.

## Differential effects of endocytic accessory proteins (EAPs) on CCP dynamics revealed by DASC

Equipped with DASC as a robust and validated tool to distinguish *bona fide* CCPs from ACs and to quantitatively measure early stages of CME, we next tested its ability to analyze and phenotypically distinguish EAP function. For this we chose a subset of eleven EAPs previously implicated in early stages of CCP initiation and maturation (*Cocucci et al., 2012*; *Henne et al., 2010*; *Ma et al., 2016*; *Umasankar et al., 2014*; *Ritter et al., 2013*; *Beacham et al., 2018*; *Wang et al., 2016*; *Srinivasan et al., 2018*; *Daste et al., 2017*; *Lo et al., 2017*; *Hawryluk et al., 2006*; *Boucrot et al., 2012*), which are uniquely captured by DASC. Our measurements allow us to segment the early dynamics in CME into discrete stages (*Figure 4A*), including stage 1: initiation, measured by *CS initiation rate (CS init. in $min^{-1}\mu m^{-2}$)*, and stage 2: stabilization, quantified by *CCP%*, which is a measure of the efficiency of nascent CCP stabilization (*Figure 4A*). Combining stage 1 and 2 measurements, we calculated *CCP rate ($min^{-1}\mu m^{-2}$)*, that is the number of CCPs appearing per unit time normalized by cell area (see Materials and methods for a detailed definition and computation of the three metrics). We further measured the lifetime distribution of CCPs, which reflects CCP maturation (stage 3). Finally, we also measured the efficiency of transferrin receptor (*TfR*) uptake, *TfReff*, using traditional bulk measurement of internalized *TfRs* as a percentage of their total surface levels, which is not

**Table 1.** Quantitative summary of EAP experiments.

| EAP | Movie number | Initiation | Stabilization | Initiation +Stabilization | Maturation | Biochemical measurements of CME | |
|---|---|---|---|---|---|---|---|
| siRNA or mutant | $n_{siCtrl}$; $n_{siEAP}$ | CS initiation rate | CCP% | CCP rate | Median lifetime of CCP | *TfRint* (internal) | *TfReff* (internal/surface bound) |
| α-PIP2 | 19, 20 | ↑36%*** | ↓27%*** | ↑27%** | ↓* | – | – |
| CALM | 20, 19 | ↓38%*** | ↓30%*** | ↓67%*** | ↑25%*** | ↑21%*** | ↓64%*** |
| epsin1 | 23, 22 | → | ↓30%*** | ↓24%** | → | ↓37%*** | → |
| Eps15 | –, 23 | ↓21%*** | ↓31%*** | ↓46%*** | ↓12%** | → | ↑22%** |
| Eps15R | –, 24 | ↓19%** | ↓17%** | ↓35%*** | → | ↓13%*** | → |
| FCHO1 | 24, 24 | → | → | → | ↑10%* | ↓30%*** | → |
| FCHO2 | –, 24 | → | ↓12%** | → | → | ↓22%*** | ↓34%*** |
| ITSN1 | 20, 19 | → | ↓33%*** | ↓30%** | → | ↓22%*** | ↓9%* |
| ITSN2 | –, 22 | ↓13%* | ↓19%*** | ↓37%*** | ↑16%*** | ↓31%*** | ↓21%*** |
| NECAP1 | 22, 21 | → | ↓26%*** | ↓39%*** | ↑20%* | → | ↓24%** |
| NECAP2 | –, 21 | → | → | → | → | → | ↓13%** |
| SNX9 | 24, 24 | ↓33%*** | ↓38%*** | ↓64%*** | ↑54%*** | ↑21%*** | ↓57%*** |

↑=increase; ↓=decrease; →=no significant change, p-value>0.05; *** p-value<0.001; ** p-value<0.01; * p-value<0.05 (statistical tests explained in Materials and methods). P-values and percentage changes are mean values obtained from 300 bootstraps comparing between KD conditions and bootstrapped siControl.

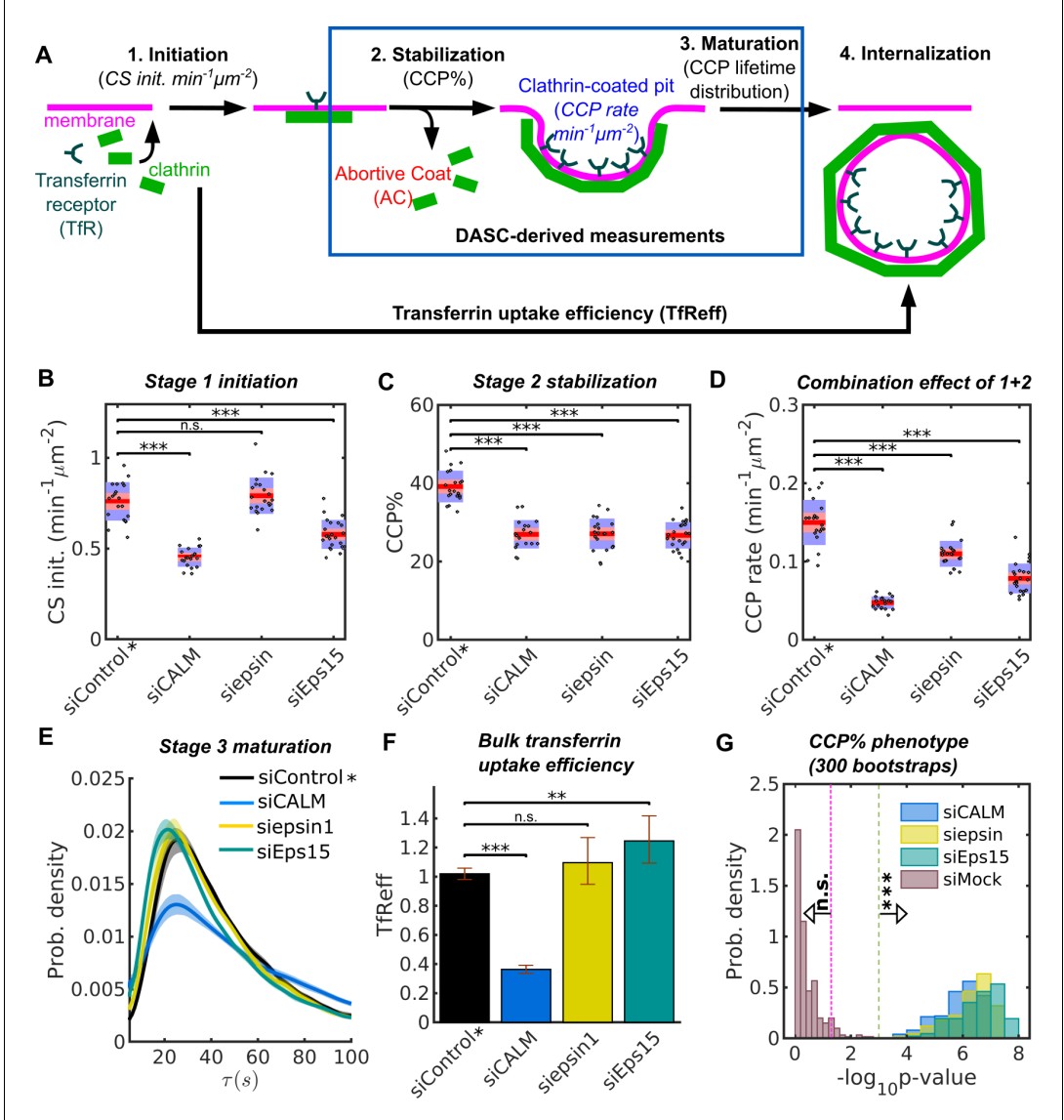

**Figure 4.** Stage specific phenotypes detected by DASC compared to transferrin uptake measurement. (**A**) Schematic of 4 stages of CME: CS initiation, CCP stabilization, CCP maturation and CCV internalization. Stage 1-3 are quantified by CS initiation rate (CS init. in $min^{-1}\mu m^{-2}$), CCP% and CCP lifetime distribution. Bulk assays for transferrin receptor uptake (*TfReff*) measure CCV formation are not stage specific. CCP rate ($min^{-1}\mu m^{-2}$) measures the combination of initiation and stabilization. Effects of siRNA knockdown of CALM, epsin1 and Eps15 on (**B**) CS initiation rate, (**C**) CCP%, (**D**) CCP rate, (**E**) CCP lifetime distribution and (**F**) *TfReff* (internalized over surface bound transferrin receptors, error bars as 95% confidence interval and statistical significance explained in Materials and methods). 20 bootstrapped siControl movies from all experimental days as siControl* are compared to the specific siRNA conditions to obtained the DASC determined phenotypes in (**B–D**). (**G**) Probability distribution of $-log_{10}p$ of CCP% for 300 bootstraps. The 3 conditions in (**B–D**) plus siMock (20 more of bootstrapped siControl movies) are shown. Veridical lines indicates location of n.s. ($p>0.05$) and *** ($p<0.001$) significance. See details on data pooling in Materials and methods.

The online version of this article includes the following figure supplement(s) for figure 4:

**Figure supplement 1.** Measurements of transferrin receptor uptake and siRNA knockdown efficiency.

stage specific but reflects the overall process of CME (see *Figure 4A*, *Figure 4—figure supplement 1A–C* and Materials and methods).

The KD effects of these EAPs relative to siControl (i.e. cells treated with a non-specific siRNA) were evaluated by percentage difference and statistical significance in DASC measurements, as well as *TfReff*. KD efficiency is shown in *Figure 4—figure supplement 1D*. In order to compare the magnitude of the KD effects relative to potential day-to-day viability, we pooled all of the siControl

movies (133 in total collected on multiple days) and randomly selected 20 from the pool to create a bootstrapped condition of control movies (herein referred to simply as siControl*). Then we compared all the KD conditions (~20 movies each, collected on the same day) to the bootstrapped siControl*. In addition, we randomly selected another 20 control movies from the pool to mimic the null effect, named siMock. This bootstrap process was then repeated 300 times to obtain reliable phenotype evaluation (see Materials and methods and materials for more details).

Three examples from a single bootstrap comparing cells treated with EAP-specific siRNA vs non-targeting siRNA on stage 1-initiation, stage 2-stabilization, stage 1 plus 2 and stage 3-maturation are shown in *Figure 4B–E*. Additionally, the result of *TfReff* for these conditions is shown in *Figure 4F*. KD of CALM dramatically decreased initiation (*Figure 4B*), stabilization (*Figure 4C*) and *TfReff* (*Figure 4F*), and also significantly altered the lifetime distribution of CCPs (*Figure 4E*). These changes included increases in both short- and long-lived CCPs, indicative of a role for CALM in multiple aspects of CCP maturation. Conversely, KD of epsin1 selectively perturbed CCP stabilization without affecting initiation, CCP lifetime or *TfReff* (*Figure 4B–F*). Initiation and stabilization were significantly decreased upon KD of Eps15, while CCP lifetime was not significantly affected; on the other hand, *TfReff* was slightly increased (*Figure 4F*), suggesting a compensatory effect. Together, these examples show consistent and significant defects in early stage(s) caused by the three EAPs, despite their differential and less interpretable effects on the efficiency of transferrin receptor uptake.

300 repetitions of the bootstrap confirm the significance of the above observations. As an example in *Figure 4G*, the p-values of the 300 bootstraps for effects on CCP% after siRNA KD of CALM, epsin and Eps15 all exceed the significance of $p<0.001$, whereas the p-values of the siMock condition (reflecting day-to-day viability) are almost exclusively insignificant. Thus, the DASC phenotypes caused by the missing EAP functions (*Figure 4B-F*) are far more significant than day-to-day variability and by using the pooling-bootstrapping method, movies acquired on different days can be directly compared.

## Multiple EAPs affect early stages in CCP stabilization and maturation

The above analysis was extended to all 11 EAPs. For a given condition, as above, we bootstrapped and obtained 300 percentage difference ($\Delta_r$) and p-values ($p$) relative to siControl* in every DASC variable. For example, this computation results in relatively large standard deviations of $p$ for siepsin (*Figure 5A*, the yellow box plot with dotted edges, corresponding to the y axis) but extremely narrow 95% confidence intervals (small black region in the middle of the box). These data indicate that the evaluation of the mean p-values $\bar{p} = \langle p \rangle_{300\ bootstrap}$ is highly reliable. Similarly, the mean $\Delta_r$, $\overline{\Delta_r} = \langle \Delta_r \rangle_{300\ bootstrap}$, corresponding to the x axis and indicated by the horizontal box plot with magenta edge is also reliable. $\bar{p}$ and $\overline{\Delta_r}$ for other conditions show a similar pattern. We thus use $\overline{\Delta_r}$ and $\bar{p}$ to define all the phenotypes (*Figure 5—figure supplement 1A* and *Table 1*). From this we observe that FCHO1/2, ITSN1/2, NECAP1 and Eps15/15R (*Ma et al., 2016*) selectively altered CCP initiation and/or stabilization without affecting CCP maturation rates, and have only relatively mild effects on the efficiency of *TfR* uptake (*Table 1*). In sum, DASC is a statistically reliable method to detect and distinguish phenotypes caused by KD of individual EAPs, thus enabling their effects on specific stage(s) of CCP dynamics to be mechanistically dissected.

## Phenotypes detected by DASC analysis are not reflected by biochemical measurements of CME efficiency

We next evaluated the sensitivity of DASC and its relation to bulk biochemical measurement of transferrin uptake (*TfReff*), the commonly used assessment of CME efficiency. Strikingly, KD of most EAPs significantly reduced *CCP rate* (by over 30%) but caused less and/or uncorrelated shifts in *TfReff* (*Figure 4D, F* and *Table 1*). To further explore this observation, for each KD condition we replotted $\overline{\Delta_r}$ for each of the DASC phenotypes, as well as $\Delta_r TfReff$ in a colored 'heat' map (*Figure 5B*). We also included $\Delta_r$ of transferrin receptor internalization ($\Delta_r TfRint$), which is independent of potential changes in surface levels of the recycling *TfR*, as this parameter is often measured by FACS or fluorescence imaging. As is evident from this plot, DASC-detected changes in early stages of CME, that is $\overline{\Delta_r}CCP\%$ and especially $\overline{\Delta_r}CCP\ rate$, were typically more severe than effects measured by cargo uptake, that is $\Delta_r TfRint$ and $\Delta_r TfReff$. Notably, few of the early acting EAPs

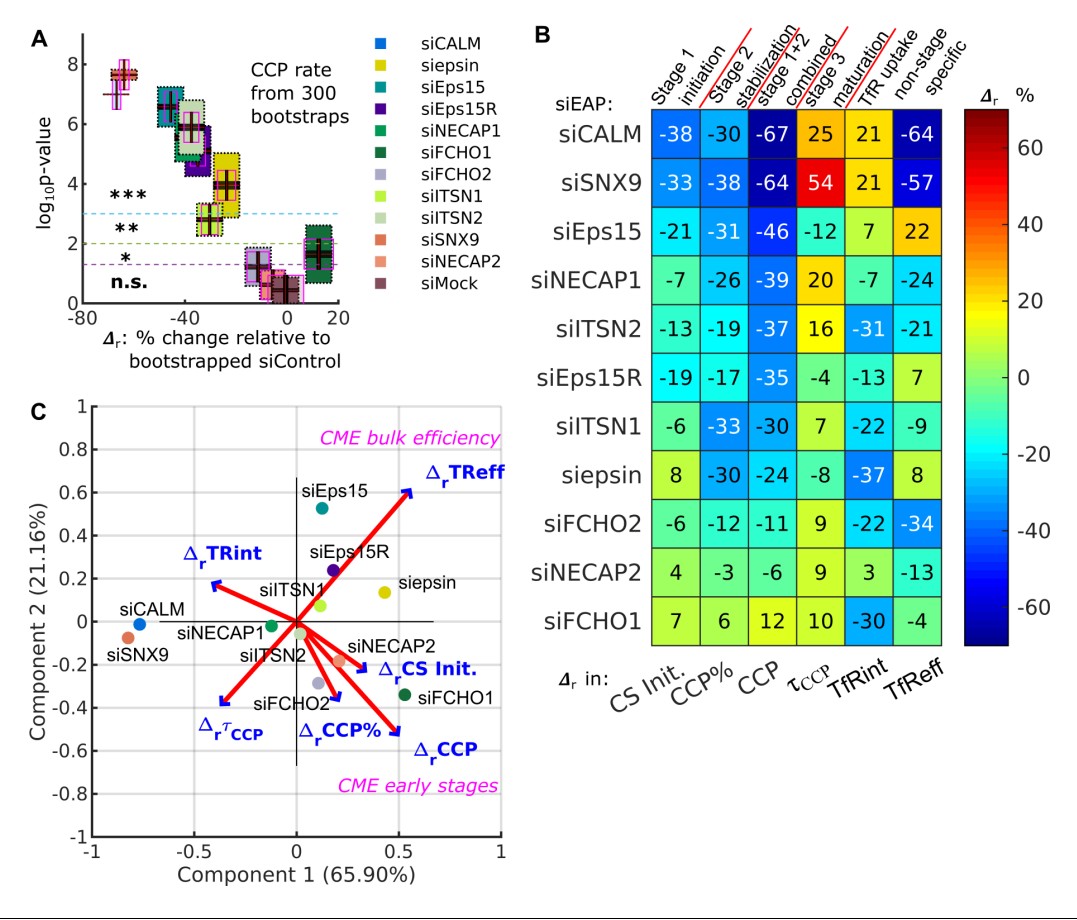

**Figure 5.** DASC is a sensitive measure of stage-specific defects in CME not detected by bulk measurement of transferrin uptake. (A) $\Delta_r$ (percentage difference) and p-values in CCP rate by comparing 11 EAP KD conditions plus siMock (Fig. 4G) to bootstrapped siControl*. Both quantities are obtained 300 times through bootstrapping. Colored boxes with black dotted edges, correspond to the p-values in the vertical axis; boxes outlined by magenta edges, correspond to $\Delta_r$ values in the horizontal axis. Red lines represent means ($\bar{p}$ and $\overline{\Delta_r}$) the black regions represent 95% confidence intervals and 1 standard deviations as colored blocks of the 300 bootstrap results. Legend shows the color for each condition. Significance level is indicated by the dashed horizontal lines. (B) Summary of phenotypes of the 11 conditions evaluated by the $\overline{\Delta_r}$ in CS initiation rate (CS init.), CCP%, CCP rate, and CCP median lifetime ($\tau_{CCP}$), and $\Delta_r$ in transferrin receptor uptake: internalized and efficiency (TfRint and TfReff) relative to control. EAP KD sorted from low $\overline{\Delta_r}$ CCP rate to high. (C) Principle component analysis (PCA). Projection of 6 variable values from 11 conditions in (B) into principle component space. First and second component (Component 1 and 2) account for 65.90% and 21.16% of total variance, respectively. Projection of original variable axes presented as red vectors with blue arrows.

The online version of this article includes the following figure supplement(s) for figure 5:

**Figure supplement 1.** Shifts in CS initiation rate and CCP% upon EAP knockdowns and correlations between all the variables.

affected CCP median lifetime ($\overline{\Delta_r}\tau_{CCP}$) and thus later stages of CCP maturation. However, those that did (i.e. CALM, SNX9, NECAP1 and ITSN2) also exhibited the strongest effects on cargo uptake.

To further illustrate the distinguishing power of DASC vs. biochemical CME measurements, we reduced the dimensionality of the extracted features for the entire collection of EAP KDs using a principal component analysis (PCA) (implemented in Matlab's function *pca*). The original data (*Figure 5B*) contained 11 observations (11 EAP KDs) of 6 variables/dimensions (6 relative changes). First, the original observations were re-centered, rescaled and projected into a new 2-dimensional PCA space, spanned by Component 1 and Component 2, which are linear combinations of the original 6 dimensions (*Figure 5C*, implemented in Matlab's function *biplot*). The variance of the original data was largely maintained (>85%) in this new space, shown by Component 1 (65.90% of total variance) and Component 2 (21.16% of total variance). Hence, the dimensionality reduction to a 2D space caused no substantial information loss. This analysis reveals that $\overline{\Delta_r}CCP\ rate$ was almost

perpendicular to $\Delta_r TfReff$. This striking lack of correlation indicates that manipulations of early CME stages have almost no direct influence on the bulk efficiency of cargo uptake. We supplemented the PCA with a correlation map (*Figure 5—figure supplement 1B*). Indeed, $\overline{\Delta_r CCP\ rate}$ among other early variables shows little correlation to $\Delta_r TfReff$, whereas changes in median lifetime ($\overline{\Delta_r \tau}_{CCP}$) showed the strongest correlation to $\Delta_r TfReff$. These comparisons highlight the value of DASC for increased sensitivity and greater phenotypic resolution over bulk biochemical measurements of cargo uptake, which can often obscure effects of EAP KD due to the resilience of CME.

## Discussion

### Unbiased classification of functionally and physically distinct abortive coats and bona fide CCPs by DASC

Live cell imaging has revealed remarkable heterogeneity in the intensities and lifetimes of eGFP-CLCa-labeled CCPs in mammalian cells, even amongst productive pits that internalize cargo (*Mettlen and Danuser, 2014*). Consequently, it has been challenging based on these two parameters to track CME progression in general and to objectively distinguish abortive coats (ACs) from *bona fide* CCPs, in particular, in order to define the contributions of the many EAPs to CCV formation. Here, we introduce DAS as a new feature space for describing CS dynamic behaviors, in which ACs and *bona fide* CCPs are accurately resolved. The DAS features exploit fluctuations in the inherently noisy intensity traces of individual CSs. The associated software pipeline, DASC, reliably separates dynamically, structurally and functionally distinct CS subpopulations without imposing any prior assumptions or the need for additional markers. While we have applied this method to the classification of CSs during CME, the DASC framework should be applicable to any localized macromolecular assembly process for which statistically sufficient traces of the addition and exchange of subunits can be provided.

Characterization of the DASC-resolved AC and CCP subpopulations shows that ACs: i) have much lower average intensities than CCPs, ii) have much shorter average lifetimes than CCPs, iii) exhibit unregulated exponentially decaying lifetime distributions, as compared to the Rayleigh distributed CCP lifetimes, iv) contain fewer AP2 complexes than CCPs, v) recruit both clathrin and AP2 at a much slower rate than CCPs, and vi) acquire less curvature than CCPs. All of these features reproduce the properties of abortive coats inferred from previous studies (*Aguet et al., 2013*; *Loerke et al., 2009*), thus both validating the robustness of DASC for distinguishing ACs from *bona fide* CCPs and providing unambiguous mechanistic insight into physical and functional properties required to stabilize nascent CCPs. Importantly, however, the distributions of each of these distinguishing properties have strong overlap between ACs and CCPs, preventing the use of any single or combined feature set as a marker for AC vs CCP classification. DASC is thus a powerful new tool for stratifying individual CSs into these groups and is uniquely applicable to single channel imaging.

### The applicability and application of DASC

DASC exploits spontaneous fluctuations in the assembly of the clathrin coat. Implicitly, the DAS algorithm relies on the assumption that these structurally-driven fluctuations dominate other image fluctuations, in particular those associated with image acquisition. To satisfy this condition, DASC imposes certain requirements on the processed movies. First, movies need to be sampled near the frequency of clathrin exchange with the assembling coat. Several studies have documented a unit turnover at the scale of ~2s, based on fluorescence recovery after photobleaching (FRAP) measurements (*Avinoam et al., 2015*; *Wu et al., 2001*; *Loerke et al., 2005*). Accordingly, all our movies were acquired at a frame rate of 1 s$^{-1}$. Second, for computing the 1) disassembly risk function $D(i, t)$, which needs statistically sufficient intensity numbers; and 2) p-values of DASC variables, which need sufficient movies to capture movie-to-movie variation, it is necessary to aggregate data sets per molecular condition that encompass $\geq 200,000$ intensity traces from ~20 movies. In addition, the movies must provide sufficient signal-to-noise ratio (SNR) for the detection of the dim, AC-related structures. As discussed before (*Aguet et al., 2013*), even with the most efficient camera equipment for image capture, cells expressing endogenous levels of fluorescent clathrin generate too dim a signal for the analysis of early coat assembly (*Figure 2—figure supplement 1*). Finally, movies must be long enough to capture the vast majority of long-lived CSs without truncation artifacts. In our case,

we filmed cells for 7.5 minutes. Note that the long acquisition could cause photobleaching. Therefore, to avoid photobleaching but maintain sufficient SNR, the laser power must be carefully tuned, as described in Materials and methods.

## Identifying molecular markers that distinguish ACs from CCPs

Numerous studies have attempted to define molecular requirements and markers that determine CCP stabilization and maturation. However, in the absence of unbiased means to distinguish between abortive and productive CCPs from clathrin signals alone, attempts have been made to identify secondary markers that unambiguously identify productive events. Besides complicating data collection, most commonly used markers remain ambiguous. For example, while a burst of Dyn2-eGFP recruitment often occurs at later stages of CCP maturation, accompanying membrane fission, the timing and intensity of these bursts vary considerably (*Taylor et al., 2011*; *Cocucci et al., 2014*); and in some cases are undetectable (*Taylor et al., 2011*). Although CCPs lacking detectable Dyn2 were indeed, on average, shorter-lived than those bearing Dyn2, the lifetime distributions of these two populations significantly overlap (*Aguet et al., 2013*; *Ehrlich et al., 2004*). Moreover, Dyn2 has been shown to be recruited at earlier stages in parallel to CCP growth (*Aguet et al., 2013*; *Ehrlich et al., 2004*; *Taylor et al., 2012*; *Cocucci et al., 2014*), and has been suggested to function in regulating the turnover of abortive pits (*Loerke et al., 2009*). Thus, the recruitment of Dyn2 does not unambiguously distinguish ACs from CCPs. Currently, the only unambiguous marker of a productive fission event is the internalization of pH sensitive-cargo (*Taylor et al., 2011*), with the obvious drawback of requiring a complex perfusion apparatus and 3-color imaging.

Others have suggested that cargo recruitment is a critical determinant of CCP maturation (*Loerke et al., 2009*; *Ehrlich et al., 2004*), and indeed CCPs with higher concentrations of *TfR* tend to have longer lifetimes than those bearing fewer *TfR* (*Liu et al., 2010*), although again there is significant overlap. However, because AP2-cargo interactions are essential for activation of AP2 complexes (*Jackson et al., 2010*) and for CCP initiation and stabilization (*Kadlecova et al., 2017*), cargo loading is also not a definitive molecular marker for ACs vs CCPs. Indeed, *TfR* intensity has been shown to correlate precisely with clathrin intensity in both small and large CCPs (*Taylor et al., 2011*). Interestingly, we have shown that the rate and extent of AP2 recruitment to nascent CCPs, which might be a surrogate for cargo recruitment, is a distinguishing feature of ACs and CCPs. Further studies will be needed to define other potential markers that either qualitatively, or more likely quantitatively, co-segregate with ACs or CCPs. We speculate that combinations of interdependent molecules rather than a single molecular event will be needed to drive and therefore define productive endocytic events.

## Are all DASC-defined CCPs productive?

Our earliest studies defined statistically distinct subpopulations of CSs, which we termed early abortive, late abortive and productive (*Loerke et al., 2009*). Later, as the sensitivity of detection increased, we used cmeAnalysis to define subthreshold CSs based on their failure to grow past a defined intensity threshold, likely encompassing the previously identified early abortive events (*Aguet et al., 2013*). The remaining 'bona fide' CCPs, which likely encompassed both 'late abortive' and 'productive' CCPs, exhibited a broad, Rayleigh-type distribution of life-times reflective of a regulated process. In support of this, we found a remarkable shift in the distribution of 'bona fide' CCPs towards short-lived exponentially decaying lifetimes when the recruitment of multiple EAPs to AP2 adaptors was perturbed by deletion of the α-appendage domain (*Aguet et al., 2013*). Interestingly, these short-lived CCPs also failed to gain curvature. We interpreted the abortive turn-over of these flat, but full-sized CCPs, as reflecting the existence of a regulatory endocytic check point. However, we have since reported other conditions that result in a shift in distribution of 'bona fide' CCPs to more rapid and exponentially decaying lifetimes, for example after acute activation of Dyn1 (*Reis et al., 2015*) or the complete replacement of CLCa with CLCb (*Chen et al., 2017*). Both of these effects on lifetime distribution were interpreted as an increase in the rate of maturation of productive CCPs (*Reis et al., 2015*; *Chen et al., 2017*). These examples highlight the current subjectivity of analyses, by us and others, and the importance of DASC for the unbiased discrimination of ACs and CCPs.

We expect that the CCPs identified by DASC also encompass a fraction of late abortive CCPs whose numbers will increase when the endocytic checkpoint is triggered. Future studies are needed to identify these factors, and importantly to verify their association with kinetically distinct subpopulations through dual channel imaging and computational segregation of DASC-defined CCPs. Given the robustness of CME and the functional redundancy of EAPs, we underline again that efficient CCP maturation is likely driven by a complex mixture of conditionally coupled EAPs and that the compositional differences between productive and late abortive CCPs thus will be quantitative (i.e. determined by conditional probability distributions) rather than qualitative (i.e. all or none).

## DASC reveals functions of early acting EAPs not detectable by bulk biochemical measurements

The power of DASC to unambiguously classify ACs from CCPs was used to analyze early acting EAPs. We could assign their differential functions to specific stages of CCV formation even when single isoforms were individually depleted and bulk rates of cargo uptake were not or only mildly affected. Thus, DASC enables sensitive, phenotypic assignment of individual EAPs to discrete stages of CME. The poor/lack of correlation between CCP rate and transferrin uptake suggests the existence of compensatory mechanisms (*Aguet et al., 2013*; *Chen et al., 2017*) and/or molecular redundancy (*Kirchhausen et al., 2014*) that could account for restoring or maintaining efficient cargo uptake. These resilient aspects of CME against the effects of KD of individual components of the endocytic machinery are also evident in the inability of multiple genome-wide screens based on ligand internalization assays to detect EAPs (*Kozik et al., 2013*; *Bassik et al., 2013*; *Gulbranson et al., 2019*; *Collinet et al., 2010*). Thus, DASC will be a critical tool for future studies aimed at identifying possible compensatory mechanisms able to restore transferrin receptor internalization. In particular, we and others (*Aguet et al., 2013*; *Loerke et al., 2009*; *Ehrlich et al., 2004*; *Chen et al., 2019*) have proposed that CCP maturation is controlled by an endocytic checkpoint. While the existence of such a checkpoint was inferred from statistical analysis of CS lifetimes and associated considerations of the maturation kinetics (*Aguet et al., 2013*; *Loerke et al., 2009*), the molecular nature of this checkpoint has remained elusive, in part because of the complexities in characterizing multi-component molecular machinery. DASC's single-marker classification of ACs and CCPs opens an opportunity to identify the molecular conditions of such a checkpoint through multi-channel imaging of EAPs, especially of early EAPs.

We report a strong effect on the efficiency of *TfR* uptake in cells depleted of CALM and SNX9, whereas others have reported only minor or no effects (*Xiao et al., 2012*; *Huang et al., 2004*; *Posor et al., 2013*; *Bendris et al., 2016*). These differences may reflect cell type specific expression levels and/or activities of functionally redundant isoforms such as AP180 or SNX18 in the case of CALM and SNX9, respectively (*Posor et al., 2013*). Interestingly, the effects we observe on *TfR* uptake correlate most strongly with the effects on CCP maturation resulting from depletion of CALM, SNX9 and, to a lesser extent, other EAPs (*Figure 5—figure supplement 1*). Nonetheless, being uncorrelated to but more sensitive than *TfR* uptake assays, DASC perhaps suggests a limitation of using deterministic biochemical approaches to study mesoscopic systems that are molecularly dense and complex, like CME. As a new method towards these considerations, DASC sheds some light on the effectiveness of statistics and fluctuation-based approaches for analysis of such systems.

In summary, DASC classifies the previously unresolvable ACs and CCPs using data derived from single channel live cell TIRF imaging, thus providing an accurate measure of progression of CME through its early stages. This comprehensive, unbiased and sensitive tool enables the determination of the distinct contributions of early EAPs to clathrin recruitment and/or stabilization of nascent CCPs. The stage-specific analysis by DASC is essential to characterize the functions of EAPs that were previously masked by detection limits and incompleteness of current experimental approaches. Going forward, DASC will be essential to functionally and comprehensively characterize the roles of the complete set of >70 EAPs in CME dynamics.

# Materials and methods

**Key resources table**

| Reagent type (species) or resource | Designation | Source or reference | Identifiers | Additional information |
|---|---|---|---|---|
| Cell line (*Homo-sapiens*) | ARPE19/HPV16 eGFP_CLCa | This paper | | See Materials and methods |
| Cell line (*Homo-sapiens*) | ARPE19/HPV16 eGFP_CLCa+α-AP2-WT | This paper | | See Materials and methods |
| Cell line (*Homo-sapiens*) | ARPE19/HPV16 eGFP_CLCa+α-AP2-PIP2- | This paper | | See Materials and methods |
| Transfected construct (human) | siRNA to EPS15 | Dharmacon | CONJB-000059 | Sense sequence: AAACGGAGCUACAGAUUAUUU |
| Transfected construct (human) | siRNA to EPS15R | Dharmacon | CONJB-000061 | Sense sequence: GCACUUGGAUCGAGAUGAGUU |
| Transfected construct (human) | siRNA to epsin1 | Dharmacon | CONJB-000063 | Sense sequence: GGAAGACGCCGGAGUCAUUUU |
| Transfected construct (human) | siRNA to SNX9 (pool of two) | Dharmacon | | Sense sequence: #1: AAGCACUUUGACUGGUUAUU #2:AACAGUCGUGCUAGUUCCUCA |
| Transfected construct (human) | siRNA to FCHO1 | Santa Cruz | Sc-97726 | transfected construct (human) |
| Transfected construct (human) | siRNA to NECAP1 (stealth) | Invitrogen | HSS177973 | Sense sequence: GCUCUUUGCUCAG GCACCAGUAGAA |
| Transfected construct (human) | siRNA to NECAP2 (stealth) | Invitrogen | HSS148087 | Sense sequence: CCGGCUGAGGAUCA CUGCAAAGGGA |
| Transfected construct (human) | siRNA to CALM | Miller et al. *Cell* 2011 | | Sense sequence: ACAGTTGGCAGACAGTTTA |
| Transfected construct (human) | siRNA to FCHO2 | Santa Cruz | Sc-91916 | transfected construct (human) |
| Transfected construct (human) | siRNA to ITSN1 | Qiagen | | Sense sequence: GCAAAUGCUUGGAAGACUU |
| Transfected construct (human) | siRNA to ITSN2 | Qiagen | | Sense sequence: CGUAAAGCCCAGAAAGAAA |
| Antibody | Anti-alpha Adaptin (Mouse monoclonal, AC1-M11) | ThermoFisher Scientific | MA3-061 | WB (1:1000) |
| Antibody | Anti-PICALM antibody [EPR12177] (Rabbit monocolonal) | Abcam | ab172962 | WB (1:1000) |
| Antibody | Anti-FCHO1 antibody - C-terminal (rabbit polycolonal) | Abcam | ab229255 | WB (1:1000) |
| Antibody | Anti-FCHO2 (Rabbit polycolonal) | Novus | NBP2-32694 | WB (1:1000) |
| Antibody | Anti-ITSN1 (rabbit polycolonal) | Sigma | HPA018007 | WB (1:1000) |
| Antibody | Anti-β-Actin (Mouse monoclonal) | Sigma | A1978 | WB (1:5000) |
| Antibody | Anti-SNX9 (Rabbit polyclonal) | Sigma | HPA031410 | WB (1:1000) |
| Antibody | Anti-ITSN2 (Mouse polyclonal) | Abnova | H00050618-A01 | WB (1:1000) |

*Continued on next page*

*Continued*

| Reagent type (species) or resource | Designation | Source or reference | Identifiers | Additional information |
|---|---|---|---|---|
| Antibody | Anti-Transferrin receptor (mouse monoclonal) | in-house antibody, HTR-D65, PMID:1908470 | | generated in-house from hybridomas, recognizes the ectodomain of the transferrin receptor |
| Antibody | Anti-ESPS15 (Rabbit polyclonal) | Santa Cruz | Sc-534 | WB (1:1000) |
| Antibody | Anti-ESPS15R (Rabbit polyclonal) | in-house antibody | | WB (1:1000) |
| Antibody | Anti-NECAP1 (3585) (Rabbit polyclonal) | Ritter et al. *Biochem. Soc. Trans.,* 2004 | | WB (1:1000) |
| Antibody | Anti-NECAP2 (3148) (Rabbit polyclonal) | From Brigitte Ritter | | WB (1:1000) |
| Antibody | Anti-epsin1 (goat polyclonal) | Santa Cruz | Sc-8673 | WB (1:1000) |
| Software | DASC (integrated to *cmeAnalysis*) | https://github.com/ DanuserLab/cmeAnalysis | | |

## Computational flow of DAS analysis

1. Acquire intensity traces using cmeAnalysis (*Aguet et al., 2013*) to analyze live-cell imaging movies. From the software output, determine the total number of traces, $N_{tot}$, which includes both valid traces ($N$ entries, that is always diffraction-limited with no consecutive gaps) and invalid traces ($N_{iv}$ entries, that is not always diffraction limited, and/or contain consecutive gaps) and calculate the *CS initiation rate (CS init.)*, which equals to $N_{tot}/(A \cdot T)$, where $A$ is the cell area and $T = 451s$ is the duration of each movie. Repeat this step for control and all the experimental conditions that have been collected on the same day. It is critical that a new control be performed with each data set.

2. Include only 'valid' traces in the following DAS analysis (described below) to identify subpopulations of CSs.

3. Align each trace to its first frame, which is the first statistically significant detection (*Aguet et al., 2013*). Then, for each trace, every intensity value is rounded to its nearest integer, $i \in [1, i_{max}](a.u.)$, where $i_{max}$ is the maximal rounded intensity among all the traces acquired on the same day.

4. Calculate conditional probabilities $W_t(i^-|i)$ (*i.e.* increase in intensity from $t$ to $t+1$) and $W_t(i|i^-)$ (*i.e.* decrease in intensity from $t$ to $t+1$), $t \in [1, T]$, using the entire population of traces from the control condition:

$$W_t(i^-|i) = \frac{\rho[(i^-, t+1) \cap (i, t)]}{\rho(i, t)},$$

where $\rho(i, t)$ is the probability of traces that reach $(i, t)$, and $\rho[(i^-, t+1) \cap (i, t)]$ is the joint probability of traces that reach $(i, t)$ but also reach $(i^-, t+1)$. Conversely,

$$W_t(i|i^-) = \frac{\rho[(i, t+1) \cap (i^-, t)]}{\rho(i^-, t)}.$$

Note that large numbers of traces (>200,000), typically obtained from >20 movies per condition, are required to obtain stable values of $W_t$.

5. Calculate the function $D(i, t)$, based on *Equation 2* (see main text). Note that the $D$ function is only calculated once using control traces. The same $D$, which in essence serves as a 'standard function', will be applied to directly compare data between different conditions, if collected on the same day.

6. Convert each trace to a $D$ series by substituting its intensity at each time frame (*i.e. Equation 1*) into its $D$ function (*i.e. Equation 3*). Repeat this step for all conditions.

7. Calculate the three features $d_1$, $d_2$ and $d_3$ of every $D$ series, resulting in a $N$ by 3 data set, where $N$ is the total number of $D$ series. Repeat this step for all the conditions.

8. Make the three features numerically comparable by normalizing $d_1$, $d_2$ and $d_3$ from different conditions using means and standard deviations of the control. For any given condition, the normalized $d$ reads:

$$\bar{d}_\alpha = \frac{(d_\alpha - \mu_\alpha^{ctrl})}{\sigma_\alpha^{ctrl}}, \text{ for } \alpha = 1, 2, 3,$$

where $\mu_\alpha^{ctrl}$ is the mean of all $d_\alpha$ and $\sigma_\alpha^{ctrl}$ is the standard deviation of all $d_\alpha$ in control condition.

9. Apply the k-medoid method, using $\bar{d}_1$, $\bar{d}_2$ and $\bar{d}_3$ as features, to separate the traces from a single condition into 3 clusters, CCP, AC and OT, using Euclidean distance. k-medoids (implemented in Matlab's function *kmedoids*) is chosen for its robustness over k-means. Repeat this step for all the conditions from the same day.

10. Calculate metrics such as lifetime and maximal intensity distributions and medians, population size, etc. for all traces within the same cluster. See more details of these calculations in the following sections. Repeat this step for all the conditions.

11. Calculate the fraction of CCPs, $CCP\% = n_{CCP}/N \times 100\%$ (the efficiency of CCP stabilization), as the population of CCPs, $n_{CCP}$, divided by the entire population of valid traces, $N = n_{CCP} + n_{AC} + n_{OT}$. Similarly, calculate $AC\% = n_{AC}/N \times 100\%$. Box plots with p-values are shown for CS init. and CCP% using Matlab's exchange file function *raacampbell/sigstar* by Rob Campbell. Repeat this step for all the conditions.

12. Calculate *CCP rate* that equals to $n_{CCP}/(A \cdot T)$ as the evaluation of the combined result of initiation and stabilization.

13. Evaluate statistical significance using Wilcoxon rank sum test (implemented in Matlab's function *ranksum*).

## Statistical confidence bands of probability density functions based on bootstrapping

A new statistical analysis evaluating the variation of probability density function (pdf) is developed for the data in this paper, where movie-movie variation is considered to be the dominant source of variation. First, for a given choice of variable $x$, e.g. lifetime or maximal intensity in either CCP or AC subpopulations, $x$ values pooled from all $N_m$ movies in a certain experimental condition are obtained. To equalize the contribution from different movies, $x$ values in each movie are resampled to match the same size ($n_x$) before pooling, where $n_x$ is the median of the $N_m$ movies' CCP or AC number per movie. The pdf $p(x)$ is then computed using Matlab's function *ksdensity* (default kernel smoothing factor is applied to all pdf calculations). Next, to evaluate the movie-movie variation, the $N_m$ movies are bootstrapped to obtain $N_m$ resampled movies. $x$ values from these bootstrapped movies are pooled to compute the first bootstrapped pdf $p_{i=1}^*(x)$ using *ksdensity*, where $i$ indicates bootstrap number. Repeating this part 400 times, $p_{i=1}^*(x)$ for $i = 1 \ldots 400$ are obtained. Finally, at any given value $x$, the 95% confidence band is obtained as a lower and upper bound $[p_\downarrow(x), p^\uparrow(x)]$, where $p_\downarrow = 2.5^{th}$ percentile and $p^\uparrow = 97.5^{th}$ percentile of the 400 $p_{i=1\ldots400}^*(x)$ values. The final presentation of pdf is therefore $p(x)$ as the main curve with the confidence band defined by $p_\downarrow(x)$ and $p^\uparrow(x)$.

## Data pooling for conditions acquired on different days

For a given day ($d$) of experiments, all the intensity traces from the siControl movies over time ($t$) as $I_n(t, d)$, are first adjusted to the traces from the siControl movies acquired on a standard day as $I_m(t, sd)$, through a linear transformation $I_n'(t, d) = \alpha I_n(t, d) + \beta$ for all $n = 1 \ldots N_{tot}$. The final $\alpha'$ and $\beta'$ values are determined such that the difference between the cumulative distributions of $I_n'(t, d)$ and $I_m(t, sd)$ for all $n$ and $m$ is minimized. Then all of the traces on day $d$, including movies of siEAP cells, are transformed linearly using $\alpha'$ and $\beta'$.

Next, all of the siControl movies after intensity adjustment are pooled together. Then, to begin one bootstrap, 20 movies from the pool are randomly selected (no repeat) as the bootstrapped siControl*. Another 20 movies from the pool are randomly selected as siMock. All of the movies from the siEAP conditions (~20 movies per condition) plus the 20 siMock movies as *experimental conditions* are paired with the 20 siControl* movies as *control* to go through the DASC computational flow. This bootstrap process is repeated for 300 times. After each bootstrap, CS initiation

rate, AC%, CCP%, CCP rate and $\tau_{CCP}$ of the siEAP and siMock conditions are compared to the bootstrapped siControl* for a pair of $\Delta_r$ and $p$. Finally, 300 $\Delta_r$ and 300 $p$ values for each DASC variables (*Figure 5A* and *Figure 5—figure supplement 1*) are averaged to give $\bar{\Delta}_r$ CCP%, $\bar{p}$ of CCP%, $\bar{\Delta}_r$ CCP rate, $\bar{p}$ of CCP rate and so on.

The intensity adjustment reduces day-day viability in laser power and optical conditions. In addition, there also exists intrinsic cell physiological heterogeneity, *e.g.* difference in expression level of endocytic proteins and *etc.*, mainly caused by the variation in passage number across different days. However, the resulting day-day viability is insignificant as shown by siMock in *Figures 4G* and *5A*, and *Figure 5—figure supplement 1*.

## Normalized two-dimensional distributions

DAS plots (e.g. *Figure 2D*), calculated as $\bar{\rho}(d_1, d_2) = \rho(d_1, d_2)/\max[\rho(d_1, d_2)]$ represent the 2D probability density normalized by maximum, where $\rho(d_1, d_2)$ is the probability density in $d_1$-$d_2$ space, binned by $\Delta d_1 = 0.2$ and $\Delta d_2 = 0.5$. The normalized probability density projections of the data in the $(d_1, d_2, d_3)$ space in *Figure 2A* is computed in the same way, adding bins of $\Delta d_3 = 0.5$.

The DAS difference maps (e.g. *Figure 3B*) show the difference between the normalized 2D densities of two given conditions divided by their integrations (condition one as control),

$$\Delta\rho(cond.1, cond.2) = \frac{\bar{\rho}_{cond.2}(d_1, d_2)}{\sum_{d_1}\sum_{d_2}\bar{\rho}_{cond.2}(d_1, d_2)\Delta d_1 \Delta d_2} - \frac{\bar{\rho}_{cond.1}(d_1, d_2)}{\sum_{d_1}\sum_{d_2}\bar{\rho}_{cond.1}(d_1, d_2)\Delta d_1 \Delta d_2}.$$

## Averaged intensity and $\Delta z$ time course

For a given cohort lifetime $\tau$, the traces within lifetime range $\tau \pm 5s$ are averaged using the cohort method described in *Aguet et al. (2013)*. The average values are presented as lines, and their error (standard deviation) as bands.

Using the microscopy setup illustrated in *Figure 3—figure supplement 3A*, Epi and TIRF intensities over the lifetimes of each cohort (*Figure 3—figure supplement 3B-D*) and errors of EPI and TIRF channels are obtained, that is $I_E(t) \pm \Delta I_E(t)$ and $I_T(t) \pm \Delta I_T(t)$. Following the approach developed by *Saffarian and Kirchhausen (2008)*, we then derived the distance between the center of the CS (*) and the initial position of assembled clathrin (+) as the invagination depth $\Delta z$ (*Figure 3—figure supplement 3A*). For each cohort we calculated $\Delta z(t)/h = ln\frac{I'_E(t)}{I_T(t)}$, where the normalization factor is the characteristic depth of the TIRF field, $h = 115nm$ based on our TIRF setting, similar to *Loerke et al. (2011)*. $I'_E(t)$ defines the Epi intensity trace adjusted to match the initial growth rate of clathrin measured in the TIRF intensity trace.

$I_E$ and $I_T$ are different in linear range of intensity measurement, that is the same intensity signal may have different readings from EPI and TIRF channel. To correct for this, $I_E(t)$ is adjusted along following protocol: 1) the 1th, 3rd and 4th data points in $I_E(t)$ and $I_T(t)$ are initially considered for linear fitting to obtain $P_E(t)$ and $P_T(t)$ respectively. The 2nd data point is removed for obvious unsmoothness in $I_E$ and $I_T$ at $t = 2s$, see *Figure 4Bi*. Then, 5rd ... 10th data points are one by one added to the linear fitting to reduce fitting residue. The collection of data points and associating $P_E(t)$ and $P_T(t)$ are determined for the smallest residue. Then the initial growth rate for both channels is approximated as

$$k_E = \frac{dP_E}{dt}\Big|_{t=1},$$

$$k_T = \frac{dP_T}{dt}\Big|_{t=1},$$

and $I_E(t)$ adjusted such that the growth rate of the corrected series $I'_E(t)$ matches $k_T$, that is

$$I'_E(t) = \frac{k_T}{k_E}I_E(t) + I_0, \tag{S1}$$

and $I_0$ is an additive correction factor (see below). The averaged invagination depth is then extracted from the relation

$$I_T(t) = I_E'(t) exp\left(-\frac{\Delta z}{h}\right), \tag{S2}$$

that is

$$\frac{\Delta z(t)}{h} = ln\left[\frac{I_E'(t)}{I_T(t)}\right]. \tag{S3}$$

Considering the approximation that $\Delta z(t=1) \approx 0$, $I_0$ is obtained by substituting *Equation S1* into *Equation S3*, and then replacing $I_E$ and $I_T$ at $t = 1s$ with the corresponding fitted values from $P_E$ and $P_T$,

$$I_0 = -\frac{k_T}{k_E}P_E(t=1s) + P_T(t=1s).$$

$z(t)$ is then expressed as a function of $I_T$ and the original $I_E$ with calculated parameter values,

$$\frac{z(t)}{h} = ln\left[\frac{\frac{k_T}{k_E}I_E(t) + I_0}{I_T(t)}\right].$$

The error of $\Delta z(t)$ is obtained through error propagation for the two variables $I_E(t) \pm \Delta I_E(t)$ and $I_T(t) \pm \Delta I_T(t)$ using Matlab's exchange file function *PropError* by Brad Ridder. Note that at early and late time points, high background but weak foreground intensity prohibits accurate calculation of $I_E$ and hence $\Delta z$ (*Figure 3F* and *Figure 3—figure supplement 3*). We also detected too few ACs in the 40s cohort for robust analysis (*Figure 3—figure supplement 3A*).

## Cell culture and cell engineering

ARPE19 and ARPE-19/HPV-16 (ATCC CRL-2502) cells were obtained from ATCC and cultured in DMEM/F12 medium with 10% (v/v) FBS at 37°C under 5% CO2. ARPE-19/HPV-16 cells were infected with recombinant lentiviruses encoding eGFP-CLCa in a pMIEG3 vector, and sorted by FACS after 72 hr (*Aguet et al., 2013*). AP2 reconstitution was achieved by infecting the eGFP CLCa-expressed ARPE-19/HPV-16 cells (ARPE/HPV16 eGFP_CLCa) with retroviruses encoding siRNA resistant WT or PIP2- (K57E/Y58E) AP2 alpha subunit in a pMIEG3-mTagBFP vector and FACS sorted based on BFP intensity (*Kadlecova et al., 2017*). Western blotting was used to confirm reconstituted-protein expression and knockdown efficiency of the generated cell lines using anti-alpha-adaptin (Thermo Fisher Scientific, #AC1-M11) and anti-CALM (Abcam, #ab172962) antibodies. APRE19 cells with stable expression of mRuby2-CLCa and α-eGFP-AP2 were also generated via lenti- and retroviral transduction, respectively.

H1299 and A549 nonsmall cell lung cancer cell lines were obtained from John Minna and are from the Hamon Cancer cell Center Collection (UT Southwestern). Their identity was authenticated by DNA fingerprinting (I.e. STR analysis) using Powerplex 1.2 kit (Promega). SK-Mel-2 (SKML) human skin melanoma cells, originally purchased from ATCC and verified by STR were genome-edited to endogenously tag Dyn2 by and obtained from David Drubin (Berkeley). All cell lines were checked for mycoplasm at least annually using 'MycoScope' from GenLantis and were mycoplasm free.

## siRNA transfection

200,000 ARPE-19/HPV-16 cells were plated on each well of a 6-well plate for $\geq 3$ hr before transfection. Transfections for siRNA knockdown were assisted with Lipofectamin RNAiMAX (Life Technologies, Carlsbad, CA). Briefly, 6.5 µl of Lipofectamin RNAiMax and 5.5 µl of 20 µM siRNA were added separately into 100 µl OptiMEM and incubated separately for 5 min at room temperature. SiRNA were next mixed with lipofectamin RNAiMAX and incubated at room temperature for another 10 min before being added dropwise to the cells with fresh medium. Measurements were performed at day five after plating cells following two rounds of siRNA transfection (time gap = 24–48 hr between transfections). Western blotting confirmed that the knockdown efficiency for all proteins was over 80%. Control cells were transfected in parallel with control siRNA (siCtrl) purchased from QIAGEN (Germantown, MD).

## Transferrin receptor internalization assay

Internalization of transferrin receptor was quantified by in-cell ELISA following established protocol (*Srinivasan et al., 2018*). ARPE-19/HPV-16 cells were plated in 96 well plates (15,000 cells/well, Costar) and grown overnight. Before assay, cells were starved in PBS4+ (1X PBS buffer with addition of 0.2% bovine serum albumin, 1 mM CaCl2, 1 mM MgCl2, and 5 mM D-glucose) for 30 min at 37°C incubator with 5% CO2 and then cooled down to 4°C and supplied with 100 μl 5 μg/ml HTR-D65 (anti-TfR mAb) (*Schmid and Smythe, 1991*). Some cells were kept at 4°C for the measurement of surface-bound HTR-D65, while some cells were moved to 37°C water bath for 10 min internalization and then acid washed to remove surface-bound HTR-D65. All cells were fixed with 4% paraformaldehyde (PFA) (Electron Microscopy Sciences, PA) and penetrated with 0.1% Triton-X100 (Sigma-Aldrich). After blocking with Q-PBS (PBS, 2% BSA, 0.1% lysine, 0.01% saponin, pH 7.4) for 30 min, surface and internalized HTR-D65 was probed by HRP-conjugated goat-anti-mouse antibody (Sigma-Aldrich). Color developed after adding OPD solution (Sigma-Aldrich) and absorbance was read at 490 nm (Biotek Synergy H1 Hybrid Reader).

Statistical significance of changes in internalized and surface-bound transferrin receptors (*TfRint* and *TfRsuf*) were obtained by two-sample t-test (implemented in Matlab's function *test2*). Statistical significance of changes and 95% confidence intervals in efficiency of transferrin receptor uptake (*TfReff = TfRint/TfRsuf*) were obtained using a statistical test for ratios (*Ugrankar et al., 2015*) (implemented in a customized Matlab's function).

## Microscopy imaging and quantification

Total Internal Reflection Fluorescence (TIRF) Microscopy imaging was conducted as previously described (*Loerke et al., 2009*). Cells were grown on a gelatin-coated 22 × 22 mm glass (Corning, #2850–22) overnight and then mounted to a 25 × 75 mm cover slide (Thermo Scientific, #3050). Imaging was conducted with a 60X, 1.49-NA Apo TIRF objective (Nikon) mounted on a Ti-Eclipse inverted microscope equipped with an additional 1.8X tube lens, yielding a final magnification of 108X. Perfect focus was applied during time-lapsed imaging. For EPI-TIRF imaging, nearly simultaneous two channel (488 epifluorescence/TIRF) movies were acquired with multi-dimension acquisition (MDA). Movies were acquired at the rate of 1 frame/s. cmeAnalysis was applied for CCP detection and tracking (*Aguet et al., 2013*; *Jaqaman et al., 2008*; *Loerke et al., 2011*). Variation could arise from the heterogeneity of cover glass by itself and the gelatin-coating.

## Acknowledgements

We thank Rosa Mino for generating the α-eGFP-AP2/mRuby-CLCa expressing ARPE cells, Jenny (Qiongjing) Zou for software management and Schmid lab members for helpful discussion. This work was supported by NIH grants GM73165 to SLS and GD, MH61345 to SLS and GM067230 to GD. ZC was supported by Welch grant I-1823 to SLS.

## Additional information

### Funding

| Funder | Grant reference number | Author |
| --- | --- | --- |
| National Institutes of Health | GM73165 | Sandra L Schmid Gaudenz Danuser |
| National Institutes of Health | MH61345 | Sandra L Schmid |
| National Institutes of Health | GM067230 | Gaudenz Danuser |
| Welch Foundation | I-1823 | Sandra L Schmid |

The funders had no role in study design, data collection and interpretation, or the decision to submit the work for publication.

## Author contributions
Xinxin Wang, Conceptualization, Data curation, Software, Formal analysis, Validation, Investigation, Visualization, Methodology, Writing - original draft, Writing - review and editing; Zhiming Chen, Resources, Data curation, Validation, Investigation, Writing - original draft, Writing - review and editing; Marcel Mettlen, Resources, Project administration, Writing - review and editing; Jungsik Noh, Data curation, Software, Formal analysis, Writing - review and editing; Sandra L Schmid, Supervision, Funding acquisition, Investigation, Writing - original draft, Project administration, Writing - review and editing; Gaudenz Danuser, Conceptualization, Supervision, Funding acquisition, Investigation, Methodology, Writing - original draft, Project administration, Writing - review and editing

## Author ORCIDs
Xinxin Wang (iD) https://orcid.org/0000-0002-7996-9628
Zhiming Chen (iD) https://orcid.org/0000-0002-2423-101X
Sandra L Schmid (iD) https://orcid.org/0000-0002-1690-7024
Gaudenz Danuser (iD) https://orcid.org/0000-0001-8583-2014

## Decision letter and Author response
Decision letter https://doi.org/10.7554/eLife.53686.sa1
Author response https://doi.org/10.7554/eLife.53686.sa2

# Additional files

## Supplementary files
• Transparent reporting form

## Data availability
All raw data is made available via https://github.com/DanuserLab/cmeAnalysis (copy archived at https://github.com/elifesciences-publications/cmeAnalysis), which indicates the DOI link to the repository.

The following dataset was generated:

| Author(s) | Year | Dataset title | Dataset URL | Database and Identifier |
|-----------|------|---------------|-------------|-------------------------|
| Wang X, Chen Z, Schmid S, Danuser G, Noh J, Mettlen M | 2020 | Live-cell imaging data for DASC (disassembly asymmetry score classification) analysis of calthrin-mediated endocytosis | https://doi.org/10.35092/yhjc.12198225 | figshare, 10.35092/yhjc.12198225 |

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
