## [Decision Letter]

**Acceptance summary:**

You have now added an excellent discussion on the issues requested. In particular, you clearly summarize how well DASC can scale and translate across systems and most importantly, you nicely discuss key issues regarding the definition of ACs versus CCPs and the possible molecular mechanisms involved, which will certainly be useful to the community working in clathrin-mediated endocytosis. Therefore, we are pleased to inform you that your article, "DASC, a sensitive classifier for measuring discrete early stages in clathrin-mediated endocytosis", is now ready for publication in *eLife*.

**Decision letter after peer review:**

Thank you for submitting your article "DASC, a sensitive classifier for measuring discrete early stages in clathrin-mediated endocytosis" for consideration by *eLife*. Your article has been reviewed by three peer reviewers, one of whom is a member of our Board of Reviewing Editors, and the evaluation has been overseen by Suzanne Pfeffer as the Senior Editor. The reviewers have opted to remain anonymous.

The reviewers have discussed the reviews with one another and the Reviewing Editor has drafted this decision to help you prepare a revised submission.

The manuscript by Wang and co-workers describes a new methodology to discriminate between bona fide productive Clathrin Coated Pits (CCPs) and Abortive Clathrin Coats (ACs), which the authors refer as DASC (Disassembly Asymmetry Score Classification). The assignment to CCPs and ACs is mostly based on 3 criteria: 1. the dynamics of individual clathrin structures extracted from the presumably CCP and AC populations, which follow those previously described for the corresponding structures; 2. the observation that the predicted CCP population very significantly diminishes in an AP2 mutant unable to bind PIn(4,5)P2, which was previously shown to prevent maturation of clathrin coats; and 3. the observation that the DASC-predicted CCPs acquire curvature, whereas those extracted from the AC population do not.

This method seems to reliably discriminate between productive CCPs and ACs on single color TIRF videos. Since these structures significantly overlap in terms of their maximum intensity and life span, methods that merely use these parameters on single color videos cannot automatically assign the nature of the clathrin structures. On the other hand, the experimental settings to more directly follow the recruitment of late endocytic proteins or vesicle fission itself in two color videos, are significantly more complex. Therefore, all reviewers agreed that DASC could be a useful metric to facilitate studies on the mechanisms driving maturation of clathrin structures.

As a resource paper though, a few considerations regarding the validation of the approach and the statistical treatment of the data need to be taken into account before publication:

Two reviewers indicated that further experimental validation using a more direct measurement of maturation would strongly reinforce the work. The authors use acquisition of curvature as a read-out for maturation, but a subset of ACs seem to acquire curvature anyway (Figure 3F). Preferably cargo loading, but alternatively, either acquisition of late endocytic proteins or fission itself, could be valid parameters. The authors could use already published data from their laboratory or others, if available.

Also, it will be important to clarify a number of experimental settings as well as some of the statistical treatment of the data to better define the limits of the approach:

1) The authors do a good job of characterizing d1, d2, and d3, their prominent output measures from DASC. But the authors need to convince readers of the generalizability of this metric on 2 fronts:

a) Within-condition variability. The permutations testing for differences between% CCPs in cells from the same condition in Figure 2F-G is excellent, but only captures a small component of the potential variability between cells. Figure 4B-D shows clear differences in the siControl group for each experiment, suggesting variability in the downstream output measures. What is the source of this variability? The day of the experiment? The batch of transfection? The quality of recorded images? As this is an exacting quantitative measure, understanding the sources of variability is necessary. Perhaps the authors could replicate the permutation test, but comparing across days or across transfections? Because the authors want to make this a resource, understanding variability would help to set the limits on how experiments within a dataset could be compared by future users of the resource.

b) Stability of Output Metric Distributions. The distributions for d1-3 shown in Figure 1D-F show striking differences between the proposed classes. But what do these distributions look like between cells? Are the distributions shown in this figure summaries for a single cell? Averaged across a population? Calculating CI on these distributions, or at least showing examples from different cells, would go a long way to demonstrate the variability that future experimenters using this metric should expect to see.

2) Perhaps even more important is the threshold used in the k-medoids clustering. While the OT population appears easily separable, the AC vs. CCP populations have clear overlap. Because of how clustering works, you'll always get 3 populations if you tell the algorithm to return 3 populations, regardless of how well separated these populations are. It is very clear that the peaks of these populations are well separated (as demonstrated in Figure 2D), but what of the events that lie at the boundary between the two distributions? Since differentiating ACs from CCPs is an essential task DASC will be used for, it would be very useful to see:

a) How do events at the edge of the two clusters look? Plotted in intensity per time space seems the most useful representation. Are these events at the edge discernible as AC or CCPs from one another based on an additional metric beyond d1-3 that the authors could incorporate into the clustering parameter space?

b) How robust are k-medoid thresholds for these two groups between experiments? All the clustering is done on centered and scaled data, but once scaled can we easily compared those data across conditions? This is important for understanding how to compare data between different experimental groups in the same paper, and also comparing new data gathered with this tool in the future to previously collected results.

---

## [Author Response]

[…] As a resource paper though, a few considerations regarding the validation of the approach and the statistical treatment of the data need to be taken into account before publication:Two reviewers indicated that further experimental validation using a more direct measurement of maturation would strongly reinforce the work. The authors use acquisition of curvature as a read-out for maturation, but a subset of ACs seem to acquire curvature anyway (Figure 3F). Preferably cargo loading, but alternatively, either acquisition of late endocytic proteins or fission itself, could be valid parameters. The authors could use already published data from their laboratory or others, if available.

While our previous data showed highly significant differences in curvature acquisition between ACs and CCPs, we have nonetheless improved our Epi-TIRF analysis to reduce distortion of the curvature measurement caused by intensity fluctuation. The improved figures (Figure 3F and Figure 3—figure supplement 2B-D) show even clearer differences between CCPs and ACs in curvature acquisition.

Nonetheless, we have taken this criticism to heart and made several attempts to use a second marker.

a) First, we attempted to use recruitment of the fission protein Dyn2 to distinguish ACs from CCPs by imaging genome-edited A549 and SKML2 cells expressing eGFP-Dyn2 and mRuby-CLCa. In these cells, we found that short (10-20s) but high intensity Dyn2 bursts appear in both of DASC identified CCPs and ACs. This result was previously shown in our work using genome-edited SKML cells (F. Aguet et al., 2013) and recently observed again in genome-edited H1299 cell lines (S. Srinivasan et al., 2018) (see Author response image 1). Note that the apparent ratio Dyn2:Clathrin intensities is higher in short-lived CCPs than in longer-lived cohorts (Author response image 1, dashed boxes). Whether this is an artifact of application of the master/slave detection to these short-lives structures or a true reflection of a distinct early role for Dyn2 in regulating CME remains unclear.

b) These findings are consistent with numerous studies from the Schmid lab, which have revealed earlier functions for dynamin in CCP maturation. Early immunoEM studies (Damke et al., JCB 1994; Warnock et al., MBoC 1997) showed that Dyn2 is present on both flat and curved CCPs (i.e. at both early and late stages of CME). Moreover, in as yet unpublished studies, we have confirmed, using cmeAnalysis and DASC, that the effects of Dyn2 KD are predominantly seen during early stages of CME. Thus, we do not believe that Dyn2 is an unambiguous marker of abortive vs. productive events. We now cite examples of this from the literature (work from the Kirchhausen and Merrifield labs) in the Introduction.

c) As for cargo loading, we tried to image fluorescently tagged transferrin in ARPE cells; however due to low signal:noise in CCPs this approach was not feasible.

d) We also discussed in the Introduction the pitfalls of using GAK as a marker for late stages in CME. Controversy exists as to a potential role for GAK in early cargo loading (see Chen et al., 2019 vs. He et al., 2020, for example) and it is clear from Kirchhausen’s most recent study that without genome-editing the results are variable. Interestingly, we note that short-lived <20s CCPs are excluded from the cohort analyses in both He et al., 2020, and by us in Srinivasan et al., 2018, likely because of the artefact we describe in point a.

e) We would argue, based on these and our findings (Figure 3) that the extent of AP2 recruitment is the most accurate molecular predictor of ACs vs. CCPs.

The major advantage of DASC, as we have now more clearly emphasized in the text, is that it can resolve kinetically, physically and functionally distinct ACs and CCPs without the need for complex experimental set-ups, multi-channel imaging or genome-editing cells to obtain reliable 2^nd^ markers. We would argue that this is the most significant contribution to the field. Finally, as a community we are now able to molecularly dissect putative compositional differences between ACs and CCPs. We suspect that the differences will not be in a single component, but in a complex compositional mix. We also hope that with DASC we will finally be able to get at the core of an elusive checkpoint, which we have mathematically clearly defined, but are yet in search of an experimental handle – probably for the same reason: it takes several components to assemble a check point, especially in the context of a cell function as versatile as CME.

**Author response image 1. respfig1:** Evidence of early appearance of Dyn2 independent on clathrin dynamics during CME. (**A**) Short and bright burst of Dyn2 (blue box) in genome edited SKML cells expressing Dyn2-eGFP (adapted from Aguet et al., 2013). (**B**) Similar burst in H1299 cells (adapted from Srinivasan et al., 2018). DASC determined CCP and AC groups are very similar with regard to the intensity of endogenous Dyn2-eGPF in both (**C**) SKML and (**D**) A549 cells.

1) The authors do a good job of characterizing d1, d2, and d3, their prominent output measures from DASC. But the authors need to convince readers of the generalizability of this metric on 2 fronts:a) Within-condition variability. The permutations testing for differences between% CCPs in cells from the same condition in Figure 2F-G is excellent, but only captures a small component of the potential variability between cells. Figure 4B-D shows clear differences in the siControl group for each experiment, suggesting variability in the downstream output measures. What is the source of this variability? The day of the experiment? The batch of transfection? The quality of recorded images? As this is an exacting quantitative measure, understanding the sources of variability is necessary. Perhaps the authors could replicate the permutation test, but comparing across days or across transfections? Because the authors want to make this a resource, understanding variability would help to set the limits on how experiments within a dataset could be compared by future users of the resource.

We appreciate the suggestion and have introduced a pooling-bootstrapping method to compare conditions acquired on different days. See changes in main text, detailed description in Materials and methods and Figure 4G, Figure 5A and Figure 5—figure supplement 1A. We pool all the siControl videos on different days and bootstrap 20 videos from the pool to compare the control bootstrap with all the siRNA conditions. We repeated this bootstrap process 300 times and then use the mean values of the 300 percentage differences and pvalues in all DASC variables to evaluate all the 11 KD conditions. In addition, we compared these conditions to a new siMock condition, which is obtained by bootstrapping another 20 siControl videos. siMock reflects day-to-day viability in cells. We find that siMock gives insignificant differences in all DASC variables, which confirms that the DASC revealed phenotypes are not strongly affected by day-to-day variability. Nonetheless, we still discuss possible sources of day-day variability in Materials and methods.

b) Stability of Output Metric Distributions. The distributions for d1-3 shown in Figure 1D-F show striking differences between the proposed classes. But what do these distributions look like between cells? Are the distributions shown in this figure summaries for a single cell? Averaged across a population? Calculating CI on these distributions, or at least showing examples from different cells, would go a long way to demonstrate the variability that future experimenters using this metric should expect to see.

We added 95% confidence intervals (CIs) to Figure 1D-F. As described in Materials and methods, these CIs reflect same day video-to-video variability and the solid curves are the averages of all the videos acquired on the same day. Nevertheless, it is clear that the bimodal distribution is not diminished by the CIs, which suggests the robustness of the bimodality in the presence of video-to-video viability.

2) Perhaps even more important is the threshold used in the k-medoids clustering. While the OT population appears easily separable, the AC vs. CCP populations have clear overlap. Because of how clustering works, you'll always get 3 populations if you tell the algorithm to return 3 populations, regardless of how well separated these populations are. It is very clear that the peaks of these populations are well separated (as demonstrated in Figure 2D), but what of the events that lie at the boundary between the two distributions? Since differentiating ACs from CCPs is an essential task DASC will be used for, it would be very useful to see:a) How do events at the edge of the two clusters look? Plotted in intensity per time space seems the most useful representation. Are these events at the edge discernible as AC or CCPs from one another based on an additional metric beyond d1-3 that the authors could incorporate into the clustering parameter space?

1) We validated our claim for three existing clusters (k=3 in k-medoids clustering) through our analyses on HPV-RPE cells. We used the elbow method to test k=3 as the optimal choice of cluster number. We observed a clear elbow at k=3 in the distance vs k plot for this cell line. In addition, we tested another 5 cell lines and observed the k=3 elbow in most of them. A549 is the exception. We noted that in this cell line, CCP dynamics is much lower and there is an abundance of flat clathrin lattices. As a result, for the duration of our videos, many static structures exist, which distorts the DASC analysis. We discuss our confirmation of k=3 in the main text and see new Figure 2—figure supplement 1.

2) We also analyzed the ‘edges’ of the clusters (new Figure 2—figure supplement 2). We found that traces close to the boundary between the CCP and AC populations classified as CCPs have lower intensities and shorter lifetimes than average CCPs but higher intensities and longer lifetimes than average ACs. This is an expected phenomenon in clustering of continuously distributed data points. However, this ambiguity of edge traces has no effect on our ability to resolve functionally, structurally and kinetically distinct AC and CCP populations. We confirmed this by testing how much exclusion of the 10% traces closest to population boundary would change the phenotype of a molecular perturbation such as CALM KD. Focusing on the CCP%, elimination of edge traces reduces the values for both siControl and siCALM. But, the relative difference in CCP% between the two conditions remains the same. This conserved difference indicates that the originally observed phenotype that siCALM leads to reduced CCP stabilization is unchanged. Thus, the DASC, as presented, is robust, even though the AC and CCP populations are not fully separable.

b) How robust are k-medoid thresholds for these two groups between experiments? All the clustering is done on centered and scaled data, but once scaled can we easily compared those data across conditions? This is important for understanding how to compare data between different experimental groups in the same paper, and also comparing new data gathered with this tool in the future to previously collected results.

3) We apologize for not making clear that the k-mediods clusters’ thresholds are only determined using siControl videos. We have clarified this in the main text and Materials and methods. The centering and scaling is for equalizing the influence of d1, d2, d3 in the classification. The data in siRNA conditions are treated exactly the same as for the siControl data, so there is no distortion caused by centering and scaling. These thresholds are robust. As shown in Figure 4G, Figure 5A and Figure 5—figure supplement 1A, all of the DASC variables of the siMock condition present insignificant differences from siControl. The permutation test in new Figure 2G again indicates that the classification of siControl is robust. As for other siRNA conditions, the thresholds obtained from siControl are directly applied, no new threshold is calculated. Hence, there is no ambiguity potentially caused by evaluating new conditions. Confirmed by various tests for all siRNA conditions (see Figure 4G, Figure 5A and Figure 5—figure supplement 1A), we show that the phenotypes derived from these k-medoids thresholds are highly consistent. The essential reason for the robustness of the k-medoids thresholds is because of the strong modes that appear in all three directions, d1-3, of the feature space.

4) For future validation or comparison, any lab following our protocols on microscopy and cell culture should obtain consistent results. The way for other labs to compare the same experimental conditions shown in this manuscript is to 1) add their siControl videos into our siControl video pool to validate that no significant difference exists among siControl videos; 2) if 1) is confirmed, follow the pooling-bootstrapping strategy to obtain the DASC results for all of the conditions. In this process, again, all of the k-medoid thresholds will be obtained from siControl videos only. Therefore, as long as the siControl matches, there will be no ambiguity in k-medoid thresholds that obscures phenotype evaluation. To this end, with the advice from the *eLife* editors, we will upload all of our data to an appropriate cloud service so that it is accessible to all potential users.